# Closing the gap between palaeontological and neontological speciation and extinction rate estimates

Daniele Silvestro[1,2,3,4], Rachel C.M. Warnock[4,5], Alexandra Gavryushkina[6] & Tanja Stadler[4,5]

Measuring the pace at which speciation and extinction occur is fundamental to understanding the origin and evolution of biodiversity. Both the fossil record and molecular phylogenies of living species can provide independent estimates of speciation and extinction rates, but often produce strikingly divergent results. Despite its implications, the theoretical reasons for this discrepancy remain unknown. Here, we reveal a conceptual and methodological basis able to reconcile palaeontological and molecular evidence: discrepancies are driven by different implicit assumptions about the processes of speciation and species evolution in palaeontological and neontological analyses. We present the "birth-death chronospecies" model that clarifies the definition of speciation and extinction processes allowing for a coherent joint analysis of fossil and phylogenetic data. Using simulations and empirical analyses we demonstrate not only that this model explains much of the apparent incongruence between fossils and phylogenies, but that differences in rate estimates are actually informative about the prevalence of different speciation modes.

[1] Department of Biological and Environmental Sciences, University of Gothenburg, 41319 Gothenburg, Sweden. [2] Global Gothenburg Biodiversity Centre, 41319 Gothenburg, Sweden. [3] Department of Computational Biology, University of Lausanne, Lausanne 1015, Switzerland. [4] Swiss Institute of Bioinformatics (SIB), 1015 Lausanne, Switzerland. [5] Department of Biosystems Science & Engineering, Eidgenössische Technische Hochschule Zürich, 4058 Basel, Switzerland. [6] Department of Biochemistry, University of Otago, Dunedin 9054, New Zealand. These authors contributed equally: Daniele Silvestro, Rachel C. M. Warnock. Correspondence and requests for materials should be addressed to D.S. (email: daniele.silvestro@bioenv.gu.se)

Understanding the drivers of biodiversity is a major, long-standing research focus in evolutionary biology[1-5]. Since changes in taxonomic diversity reflect the combined effects of speciation and extinction processes, intense effort has gone into developing methods to quantify species diversification rates. Two primary sources of data are used to estimate speciation and extinction rates: (i) phylogenetic trees of extant taxa[6-10] and (ii) fossil occurrence data[3,11-13]. Although extant and fossil species are samples of the same underlying diversification process, a large discrepancy has been widely documented between empirical estimates of diversification rates inferred from phylogenetic versus fossil data[13-16], with few exceptions, e.g. ref. [17]. In particular, extinction rates estimated from empirical phylogenies are often much lower than expected given observed extinct diversity[18]. For example, phylogenetic estimates of diversification rates among cetaceans suggest speciation has exceeded extinction over the past 12 Myr, implying diversity has increased towards the recent[19]. In contrast, analyses of the cetacean fossil record indicate extinction has exceeded speciation over this same interval, and that the diversity of cetaceans was in fact once much higher than it is today[15,20,21].

Discrepancies remain despite the development of increasingly realistic approaches to estimating diversification rates[9,22-24], and efforts to explore the limitations of both stratigraphic and phylogenetic data[13,18,20,25,26]. Since overwhelming palaeontological evidence suggests that extinction rates close to zero are extremely improbable, some authors have questioned whether it is possible to estimate diversification rates reliably from phylogenetic data alone[15,25]. This situation is surprising since (i) methods used to estimate rates from fossils and phylogenies are based on the same underlying mathematical birth−death process theory[27,28], and (ii) both phylogenetic and palaeontological approaches perform well under simulation conditions[8,13,26]. Incongruences have been attributed to biases in the data[29], lack of statistical power[13,30,31] and violation of underlying model assumptions[20,25]. However, the magnitude of the observed discrepancies remains unexplained.

Previous studies have shown that different approaches to modelling speciation (or origination) can lead to divergent estimates of macroevolutionary parameters[32-37]. Three speciation modes (budding, bifurcation, and anagenesis) have been described in the palaeobiological literature that can leave a signature in the fossil record, without necessarily impacting the underlying phylogenetic tree[38,39] (Fig. 1). These speciation modes are assumed to result from different biological processes, and may be interpreted differently depending on the adopted species concept, e.g. morphospecies versus evolutionary species[33,40,41]. For instance, cladogenesis via bifurcation has been presented as the expected outcome of vicariance or allopatric speciation, whereas budding divergence has been interpreted as the result of peripatric speciation[38].

Here, we examine whether these alternative speciation modes, regardless of the species-generating mechanism, are responsible for driving incongruences between palaeontological and phylogenetic estimates of diversification. We unify budding, bifurcation, anagenesis, and extinction in a single 'birth−death chronospecies' (BDC) process, and explore the impact of this framework through theoretical considerations, extensive simulations and empirical analyses. We provide simple mathematical

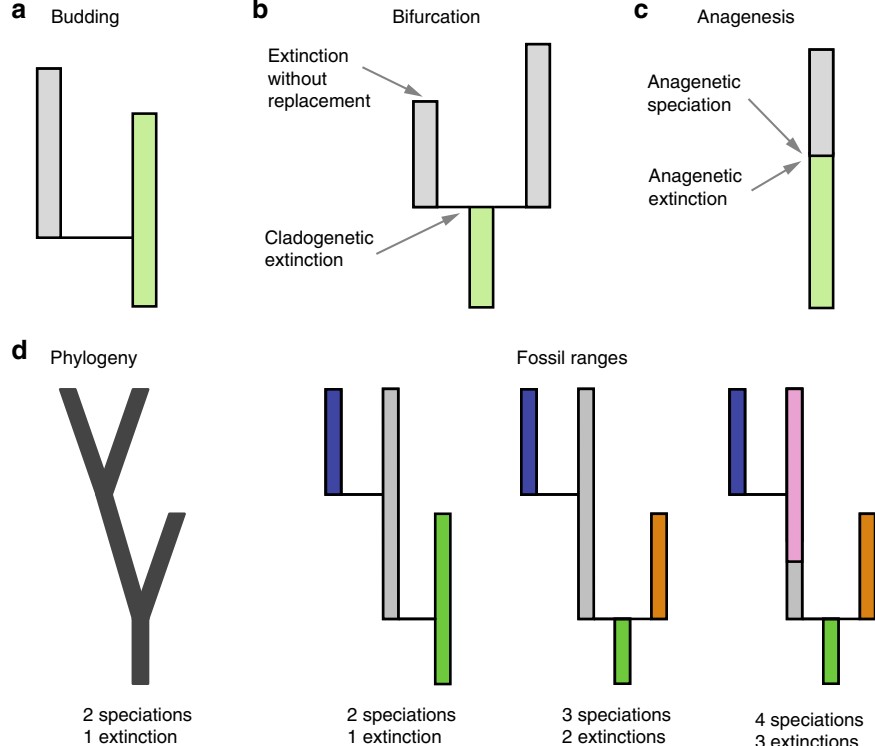

**Fig. 1** Speciation modes reflecting the difference in phylogenetic and stratigraphic interpretations of speciation and extinction rates. **a–c** Three alternative modes of speciation[39]. Each rectangle corresponds to a distinct species with green-shaded rectangles representing ancestral species. **a** Cladogenesis via budding: one new species arises, the ancestral species survives. This type of speciation happens with rate $\lambda(1-\beta)$ in the BDC model. **b** Cladogenesis via bifurcation: the ancestral species goes extinct and two new species arise with rate $\lambda\beta$. **c** Anagenesis: the ancestral species goes extinct and is replaced by one new species with rate $\lambda_a$. **d** Phylogenetic versus stratigraphic interpretations of speciation and extinction rates. The phylogeny on the left describes three possible speciation histories (three coloured trees on the right). The coloured segments represent distinct (morpho)species. Phylogenetic estimates of speciation and extinction rates, $\lambda$ and $\mu$, differ from fossil-based estimates, $\lambda^\star$ and $\mu^\star$, in two out of the three cases. The rates are only the same in the case of pure budding speciation (the first coloured tree from the left)

formulae linking the diversification rates inferred from fossils and phylogenies, taking into account alternative speciation modes, and demonstrate that different speciation modes have a large but predictable impact on incongruent diversification rate estimates. When we apply a standard birth−death model, only three out of nine empirical clades show consistent results between the fossil record and the phylogenies of extant species. However, eight out of nine empirical data sets support the BDC model, meaning fossil and phylogenetic rate estimates converge to the same diversification parameters once we account for different speciation modes. The single dataset that did not conform to the BDC model is characterised by challenging fossil taxonomy and low sampling of extant taxa. Together, our results demonstrate that there are fundamental differences in the meaning of speciation and extinction parameters inferred from fossils and phylogenies, which alone can explain much of the rate inconsistencies observed in empirical analyses. Finally, we show that the analysis of fossils and phylogenies under the BDC model can be informative about the relative importance of different modes of speciation. The BDC process unifies the definitions of speciation and extinction parameters, reflecting the differential effects of morphological evolution with and without branching events. Thus, the BDC model paves the way for a better integration of coherent models in palaeontology and phylogenetics.

## Results and Discussion

**The BDC model.** We define 'species' as an identifiable taxonomic unit (a lineage) that can persist through time, give rise to other species, and become extinct. Under this definition, the fossil record includes extinct and extant species that can be identified on the basis of morphological traits or other information (e.g. location, age, etc.), whereas phylogenetic data typically include extant representatives only, identified on the basis of phenotypic and/or genetic data. Following the terminology developed in previous work[35,38,39,42], we describe three modes that may give rise to the origination of a new species (Fig. 1), which together result in four distinct diversification processes: (1) Cladogenesis via budding: a speciation event that gives rise to one new species. The ancestral species persists and no extinction occurs; (2) Cladogenesis via bifurcation: a speciation event that gives rise to two new species, replacing the ancestral species, which becomes extinct; (3) Anagenetic speciation: evolutionary changes along a lineage that result in the origination of one new species and the replacement (extinction) of the ancestral species; (4) Extinction without replacement: a species becomes extinct without leaving any descendants. These distinct modes of speciation and extinction may reflect different evolutionary processes[38,43], in addition to variation in taxonomic practices and species definitions[33,44]. Although substantial debate remains regarding the interpretation and characterisation of the speciation process, and the extent to which some extant and extinct species may even be considered real[44,45], we consider the definition of morphospecies in the fossil record to be something of biological significance. In particular, we follow palaeontological practice by relying on an assignment of fossils to morphospecies, which in turn form species in our BDC model.

In a standard birth−death model, branching (cladogenetic) events occur at rate $\lambda > 0$, each giving rise to one additional species, and the termination of a branch (extinction) occurs with rate $\mu \geq 0$[27]. We extend this model to incorporate the possibility of alternative modes of speciation, which requires two additional parameters[35,46]. At each branching event, bifurcating speciation occurs with probability $\beta \in [0,1]$, while budding speciation occurs with probability $1 - \beta$, and anagenetic speciation occurs along each branch with rate $\lambda_a \geq 0$. We call this process with four parameters the 'birth−death chronospecies' process.

Phylogenetic estimates of diversification rates are typically obtained from dated trees of extant taxa using the reconstructed birth−death process[6,7,10]. In a phylogenetic tree, each tip is considered to be a different species, and different co-existing lineages refer to different species. However, no information about species assignment to lineages through time is available from a phylogeny, i.e. typically, we do not know if a given species prior to a branching event is identical to one of the species following the speciation event (cladogenesis via budding; Fig. 1). Existing phylogenetic approaches estimate the rate of branching and the rate of extinction, i.e. the parameters $\lambda$ and $\mu$. Under these models, extinction is assumed to be a lineage termination, and may not be associated with species replacement. Thus, phylogenetic approaches implicitly assume that all speciation events occur through cladogenesis via budding and neglect the extinction of lineages in the case of bifurcating cladogenesis and anagenetic speciation.

Palaeontological estimates of diversification rates, on the other hand, are obtained from observed stratigraphic ranges (the interval between the first and last appearances of a taxon in the fossil record) or estimated species ranges (the interval between species origination and extinction)[13,47]. Speciation and extinction rates are based on the frequencies of range origination and termination events recorded through time, and are therefore a function of the number of (morpho)species identified in the fossil record, regardless of the mechanisms that generated them. In other words, speciation rates inferred from fossil data, hereafter referred to as $\lambda^\star$, necessarily quantify all events leading to the generation of new morphospecies (with or without branching) and extinction rates (hereafter $\mu^\star$) reflect all instances leading to the disappearance of a morphospecies (with or without replacement). These parameters can be expressed as the combined contribution of speciation through budding, bifurcation or anagenesis, and extinction without replacement:

$$\lambda^* = \lambda(1 - \beta) + 2\lambda\beta + \lambda_a \qquad (1)$$

and

$$\mu^* = \lambda\beta + \lambda_a + \mu. \qquad (2)$$

In order to derive the expression for $\lambda^\star$, we note that (i) one new species may arise via budding cladogenesis (rate $\lambda(1 - \beta)$), (ii) two new species may arise via bifurcating cladogenesis (rate $2\beta\lambda$ where the 2 acknowledges that two new species arise), and (iii) one new species may arise via anagenesis (rate $\lambda_a$). Similarly, $\mu^\star$ is the sum of (i) extinction of a species due to bifurcating cladogenesis (rate $\lambda\beta$), (ii) extinction of a species due to anagenesis (rate $\lambda_a$), and (iii) extinction of a species without replacement (rate $\mu$). In summary, with fossil data sets typically including only temporal ranges of morphospecies, we can estimate overall speciation and extinction rates but, in the absence of a robust phylogenetic hypothesis or near-complete sampling, we cannot identify different speciation modes[38,48,49].

To illustrate the impact of different speciation and extinction processes on phylogenetic and palaeontological estimates of diversification rates, we performed simulations based on the BDC process under a broad range of parameters (see Methods). The initial simulations represent an ideal scenario where all species (extinct and extant) are sampled and correctly identified, and all origination, extinction and branching times are known without error. Then, we estimate $\lambda$, $\mu$ based on the phylogenies pruned of the extinct tips and $\lambda^\star$, $\mu^\star$ based on fossil data by maximum likelihood. These simulations demonstrate that we can correctly re-estimate $\lambda$, $\mu$ based on phylogenies, and $\lambda^\star$, $\mu^\star$ based on fossil data (Fig. 2).

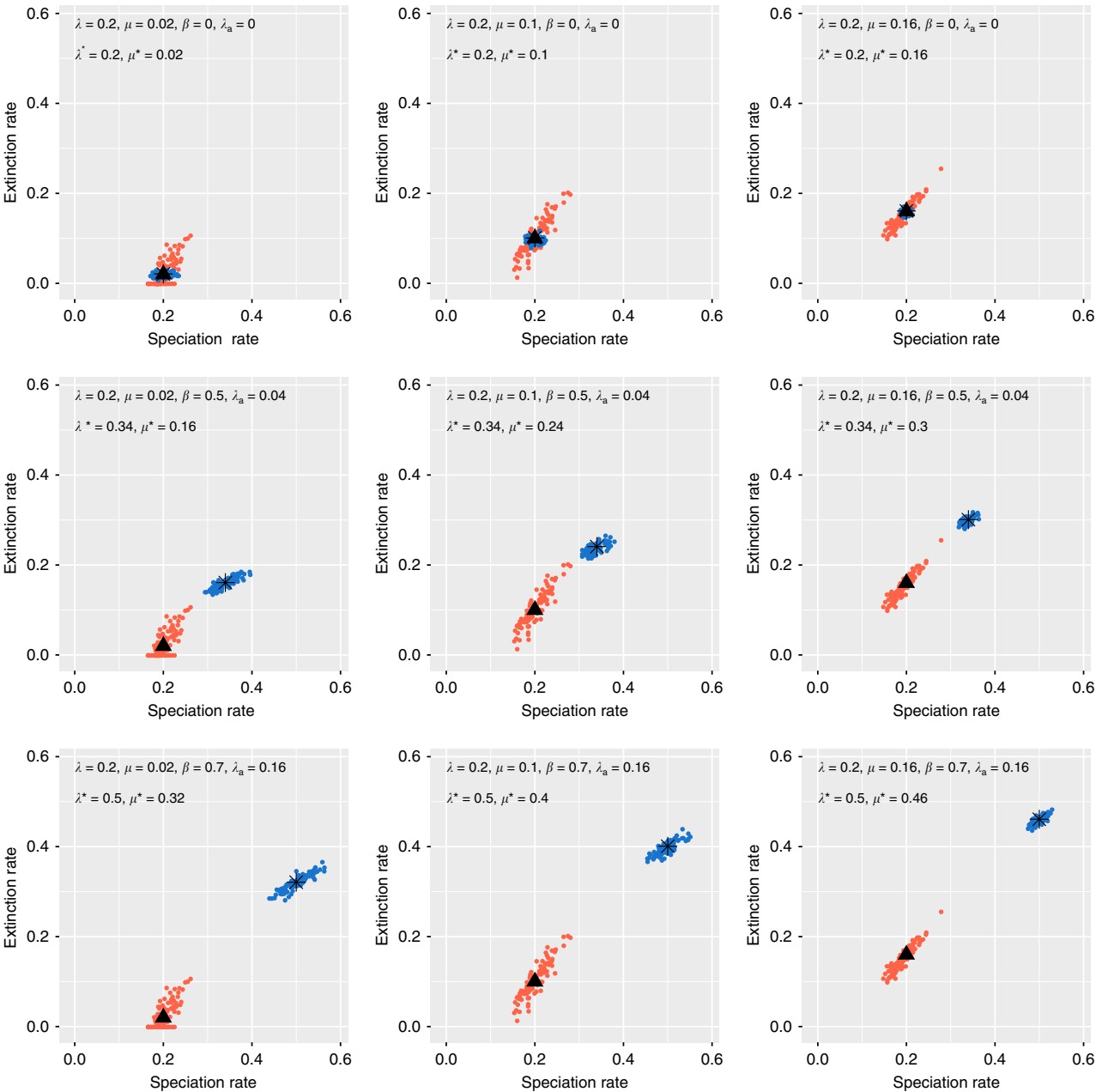

**Fig. 2** Simulations of fossil and phylogenetic data under the BDC model. We simulated different proportions of cladogenesis via budding or bifurcation, anagenetic speciation, and extinction without replacement. Phylogenetic estimates of the speciation and extinction rates ($\lambda$, $\mu$) are shown in red with black triangles representing the true values. The speciation and extinction rates ($\lambda^\star$, $\mu^\star$) estimated from fossil ranges are shown in blue with black stars representing the true values

**Mathematical exploration of the BDC model.** For given $\lambda^\star$ and $\mu^\star$ from fossils, and $\lambda$ and $\mu$ from the phylogeny, we now explore whether it is possible to obtain the unknown parameters $\lambda_a$ and $\beta$. We can write Eqs. (1) and (2) as

$$\begin{pmatrix} \lambda^* \\ \mu^* \end{pmatrix} = \begin{pmatrix} \lambda \\ \mu \end{pmatrix} + \begin{pmatrix} 1 & \lambda \\ 1 & \lambda \end{pmatrix} \begin{pmatrix} \lambda_a \\ \beta \end{pmatrix}. \quad (3)$$

Thus, we have a linear system with two unknowns ($\beta$ and $\lambda_a$) and two equations. However, since the matrix $\begin{pmatrix} 1 & \lambda \\ 1 & \lambda \end{pmatrix}$ does not have full rank, either no ($\lambda_a$, $\beta$) or infinitely many ($\lambda_a$, $\beta$) solve Eq.

(3). In particular, we have three scenarios (fulfilling our parameter constraints $\lambda > 0$, $\mu \geq 0$, $\lambda^\star > 0$, $\mu^\star \geq 0$, $\lambda_a \geq 0$, $\beta \in [0,1]$):

A. Exactly one solution, namely $\beta = 0$, $\lambda_a = 0$, exists for

$$\lambda^* - \lambda = \mu^* - \mu \text{ and } \lambda^* - \lambda = 0,$$

which is equivalent to

$$\lambda^* = \lambda, \mu^* = \mu.$$

B. Infinitely many solutions exist for

$$\lambda^* - \lambda = \mu^* - \mu \text{ and } \lambda^* - \lambda > 0,$$

namely, every $\beta \in \left[0, \min\left\{1, \frac{\lambda^*}{\lambda} - 1\right\}\right]$ with $\lambda_a = \lambda^* - \lambda - \lambda\beta$ forms a solution.

C. No solution exists if

$$\lambda^* - \lambda \neq \mu^* - \mu \text{ or } \lambda^* - \lambda < 0,$$

with the first condition coming directly from the linear system, and the second from the additional requirements of $\lambda > 0$, $0 \leq \beta \leq 1$ and $\lambda_a \geq 0$.

On this basis, we can specify three models with parameters $\lambda > 0$, $\mu \geq 0$, $\lambda^* > 0$, $\mu^* \geq 0$. The 'equal rates model' is appropriate if Scenario A above best describes the data. This corresponds to the budding speciation model where the palaeontological and phylogenetic parameters are the same, i.e. $\lambda^* = \lambda$, $\mu^* = \mu$. This model is a special case of the BDC model. These constraints are equivalent to $\lambda > 0$, $\mu \geq 0$, $\lambda^* = \lambda$, $\mu^* \geq \lambda^* - \lambda + \mu$.

The 'compatible rates model' is appropriate if Scenario B above best describes the data. This is the full BDC model, where $\lambda^*, \lambda, \mu, \mu^*$ are constrained such that $\lambda^* - \lambda = \mu^* - \mu$ and $\lambda^* \geq \lambda$; thus, this model contains the equal rates model as a special case. The constraints can be re-written as $\lambda > 0$, $\mu \geq 0$, $\lambda^* \geq \lambda$, $\mu^* = \lambda^* - \lambda + \mu$ (we note that this automatically implies that $\mu^* \geq 0$).

The 'incompatible rates model' is appropriate if Scenario C above best describes the fossil and phylogenetic data. Under this model, the parameters $\lambda$, $\lambda^*$, $\mu$, $\mu^*$ are allowed to take any value in the range $\lambda > 0$, $\mu \geq 0$, $\lambda^* > 0$, $\mu^* \geq 0$, without the constraints imposed by the BDC model (Eqs. (1) and (2)); thus differences in $\lambda$ and $\lambda^*$, as well as $\mu$ and $\mu^*$, cannot be explained by differences in speciation mode.

We determined which of the equal, compatible or incompatible rates models are supported by different simulated and empirical data sets, by estimating $\lambda$, $\mu$, $\lambda^*$, $\mu^*$ simultaneously for a given phylogeny and the corresponding fossil data (see Methods). The integration of different modes of speciation and extinction into our BDC model has substantial—in some cases unexpected—implications for the interpretation of rates estimated from phylogenetic and fossil data. Below, we highlight the most important properties of the BDC model, which emerge from the mathematical formulae above and are supported by simulations.

The first property that results from the BDC model is that phylogenetic and palaeontological speciation and extinction rate estimates will only be equal if all speciation has occurred through budding. Under the BDC model, even in an ideal scenario with fully sampled and errorless data sets, speciation and extinction rates can only be equal across phylogenetic and stratigraphic inferences if all speciation events have occurred through budding and no speciation has occurred through bifurcation or anagenesis, i.e. $\beta = 0$ and $\lambda_a = 0$ (Scenario A above, which is a special case of the BDC model). Any instance of cladogenesis via bifurcation or anagenetic speciation will alter the rates and contribute to an apparent incongruence between the parameters estimated from phylogenetic and fossil data, as demonstrated by simulations (Fig. 2, Supplementary Table 1). Both bifurcation and anagenetic speciation introduce additional speciation and extinction events without increasing the number of terminal taxa in the phylogeny. This will result in higher speciation and extinction rates estimated from fossil data compared to the rates estimated from phylogenetic trees of extant taxa. Thus, the BDC model predicts fossil-based rates to be equal to or exceed phylogeny-based rates:

$$\lambda^* \geq \lambda \text{ and } \mu^* \geq \mu, \tag{4}$$

which directly follows from the definition of $\lambda^*$ and $\mu^*$ in Eqs. (1) and (2). Another outcome of the BDC formulation is that phylogenetic estimates of extinction equal to zero (often inferred from empirical trees) do not necessarily imply that no extinction

has occurred. Indeed, $\mu = 0$ may indicate that extinctions have occurred through bifurcation events or anagenetic replacement (which translates to $\mu^* > 0$ in the fossil record). The term $\lambda^* = \lambda + \beta\lambda + \lambda_a$ (Eq. (1)) illustrates that bifurcating and anagenetic speciations contribute similarly in determining the discrepancy between stratigraphic and phylogenetic rate estimates.

A second property emerging from the BDC model is that phylogenetic and palaeontological estimates of net diversification will be equal irrespective of speciation mode. Phylogenetic and stratigraphic speciation rates, under the BDC model, differ by the same amount as the extinction rates, even if the parameters $\lambda_a$ and $\beta$ are unknown:

$$\lambda^* - \lambda = \mu^* - \mu. \tag{5}$$

Thus, while stratigraphic speciation and extinction rate estimates are likely to exceed phylogenetic speciation and extinction rate estimates, the net diversification rates are predicted to be equal:

$$\lambda^* - \mu^* = \lambda - \mu. \tag{6}$$

Our analyses of simulated data show that the model has the power to detect these equalities (Supplementary Figure 1).

Finally, the BDC model reveals that phylogenetic and stratigraphic speciation and extinction rates are informative about speciation mode. We cannot directly estimate the rates of cladogenesis via bifurcation and anagenetic speciation; however, we can still make some important statements regarding the prevalence of alternative speciation modes. First, the dependency $\lambda_a = \lambda^* - (1 + \beta)\lambda$ can be re-written as $\lambda^* - \lambda = \lambda_a + \lambda\beta$ (top row of linear system in Eq. (3)), meaning the difference between stratigraphic and phylogenetic speciation rates is the sum of anagenetic and bifurcation speciation. Second, we can provide an interval for possible values of $\lambda_a$. Assuming that a clade diversifies under the compatible rates model, we estimate parameters $\lambda > 0$, $\mu \geq 0$, $\lambda^* \geq \lambda$, and obtain the fourth parameter from $\mu^* = \lambda^* - \lambda + \mu$. Based on Eq. (3), we can obtain $\lambda_a$ given a known $\beta$ (or vice versa) from the estimated parameters $\lambda$, $\mu$, $\lambda^*$. For a known $\beta$, we obtain $\lambda_a = \lambda^* - (1 + \beta)\lambda$. Since $\beta$ ranges in the interval $\left[0, \min\left\{1, \frac{\lambda^*}{\lambda} - 1\right\}\right]$ and $\lambda_a \geq 0$, we obtain $\lambda_a \in [\max\{0, \lambda^* - 2\lambda\}, \lambda^* - \lambda]$. Thus, based on the estimated $\lambda$ and $\lambda^*$, we can provide the upper bound $\lambda^* - \lambda$ for the rate of anagenetic evolution, $\lambda_a$, and observe that the interval width is at most $\lambda$. Finally, we can assess the importance of anagenetic speciation relative to cladogenesis via budding. Since $\lambda^* - 2\lambda = \lambda_a - \lambda(1 - \beta)$, anagenetic speciation exceeds cladogenesis via budding when $\lambda^* - 2\lambda > 0$, whereas budding is more frequent compared to anagenetic speciation when $\lambda^* - 2\lambda < 0$ (Supplementary Figure 2). This property is important because it shows that the combination of fossil and phylogenetic data within the context of the BDC model, even in the absence of direct estimates of $\lambda_a$ and $\beta$, is informative about speciation mode.

**Performance and robustness of the BDC model.** If differences between phylogenetic and stratigraphic rate estimates can be explained by the BDC model, then we expect to find support for the parameter constraints $\lambda > 0$, $\mu \geq 0$, $\lambda^* \geq \lambda$, $\mu^* = \lambda^* - \lambda + \mu$, and thus for the relationships described in Eqs. (4)–(6). Analyses of simulated data sets using maximum likelihood, where we approximated the likelihood assuming that fossil and phylogenetic data are independent (see Methods), showed that the compatible rates model was correctly identified as the best-fitting model in >99% of cases (reported at the 0.99 confidence level) against the alternatives under a wide range of parameter settings

**Table 1 Performance of the BDC model across different simulation settings**

| | Simulation parameters | | | | | Freq. best model (%) | | |
|---|---|---|---|---|---|---|---|---|
| | $\lambda$ | $\mu$ | $\beta$ | $\lambda_a$ | Others | BDC: Eq | BDC: Co | In |
| Constant rates | 0.2 | 0.16 | 0 | 0 | | **95 (100)** | 4 (0) | 1 (0) |
| | 0.2 | 0.16 | 0.5 | 0.04 | | 0 (1) | **100 (99)** | 0 (0) |
| | 0.2 | 0.16 | 0.7 | 0.16 | | 0 (0) | **100 (100)** | 0 (0) |
| Incomplete sampling | 0.2 | 0.16 | 0 | 0 | $x = 0.9$ | **87 (99)** | 12 (1) | 1 (0) |
| | 0.2 | 0.16 | 0.5 | 0.04 | $x = 0.9$ | 0 (7) | **100 (93)** | 0 (0) |
| | 0.2 | 0.16 | 0.7 | 0.16 | $x = 0.9$ | 0 (0) | **100 (100)** | 0 (0) |
| | 0.2 | 0.16 | 0 | 0 | $\psi = 0.5$ | **97 (99)** | 0 (0) | 3 (1) |
| | 0.2 | 0.16 | 0.5 | 0.04 | $\psi = 0.5$ | 2 (8) | **98 (92)** | 0 (0) |
| | 0.2 | 0.16 | 0.7 | 0.16 | $\psi = 0.5$ | 0 (0) | **100 (100)** | 0 (0) |
| Rate variation | 0.2, 0.3, 0.2 | 0.16, 0, 0.16 | 0.5 | 0.04 | $t = 20, 10$ | 0 (0) | **62 (96)** | 38 (4) |
| | 0.2, 0.3, 0.2 | 0.02, 0, 0.2 | 0.5 | 0.04 | $t = 25, 15$ | 0 (0) | **59 (94)** | 41 (6) |
| | 0.3, 0.2, 0.2 | 0.02, 0.01, 0.30 | 0.5 | 0.04 | $t = 25, 15$ | 0 (0) | **2 (0.23)** | 98 (77) |
| Cryptic speciation | 0.2 | 0.16 | 0.5 | 0.04 | $\kappa = 0.1$ | 4 (22) | **96 (78)** | 0 (0) |
| | 0.2 | 0.16 | 0.5 | 0.04 | $\kappa = 0.5$ | **83 (99)** | 17 (1) | 0 (0) |
| | 0.2 | 0.16 | 0.5 | 0.04 | $\kappa = 0.9$ | 18 (38) | **0 (0)** | 82 (62) |
| | $\lambda^1$ | $\mu^1$ | $\lambda^2$ | $\mu^2$ | | | | |
| Incompatible rates | $\mathcal{U}(0.1, 1.5)$ | $\mathcal{U}(0, \lambda^1)$ | $\mathcal{U}(0.1, 1.5)$ | $\mathcal{U}(0, \lambda^2)$ | | 7 (12) | **10 (11)** | **83 (77)** |

We simulated 100 data sets (fossils and phylogenies) under each setting (for more details, see Methods) and tested the fit of models with equal (Eq), compatible (Co), and incompatible (In) rates using maximum likelihood. We calculated the frequencies with which each model was selected as the best one. We considered the best-fit model to be the simplest model not rejected by the likelihood ratio test, with a significance threshold of 0.95 (results at the 0.99 threshold are shown in parentheses). Values in bold indicate the frequency with which the correct model was selected. We indicate with $x$ the fraction of lineages randomly removed from the fossil record in the incomplete sampling simulations and with $\psi$ the Poisson sampling rate used to simulate fossil occurrences; $t$ indicates the times of rate shifts in the simulations with rate variation; $\kappa$ indicates the probability that speciation events are treated as cryptic, i.e. the new species cannot be distinguished from its parent species in the fossil record. Incompatible rates were generated by simulating independent birth–death trees for phylogenetic and stratigraphic data sets with random speciation and extinction rates

(Table 1, Supplementary Tables 1, 2). The compatible rates model was correctly rejected in >94% of cases in favour of the equal rates model when no speciation has occurred through bifurcation or anagenesis, i.e. $\beta$ and $\lambda_a = 0$. Support for the equal rates model reduces to zero in favour of the full BDC model as the frequency of bifurcating and anagenetic speciation events increases, i.e. $\beta > 0$ and $\lambda_a > 0$. Both the equal and compatible rates models were correctly rejected in favour of the incompatible rates model when fossils and phylogenies were simulated under independent processes with different rates (Table 1).

Our likelihood ratio test is robust to random, incomplete taxon sampling. Even when up to 90% of the fossil taxa are missing (Table 1, Supplementary Table 3, Supplementary Figures 3, 4), the compatible rates model was still correctly favoured over the independent rates model in 100% of the simulations. While the size of the data set certainly has an impact on the statistical power of the test, our simulations show that model testing was accurate across a wide range of realistic sampling scenarios (Supplementary Tables 3, 14). The results are also robust when fossils (and therefore ranges) are sampled under a Poisson sampling process (Supplementary Table 4, Supplementary Figures 5, 6). However, support for the compatible rates model decreases in favour of the independent rates model when sampling of fossil species is highly non-uniform (Supplementary Tables 5, 6, Supplementary Figures 7—10).

The power of our test was high (82–100%) even when the data were simulated under some scenarios that included a substantial amount of rate heterogeneity through time, which is not explicitly accounted for in our model (Supplementary Tables 7, 8, Supplementary Figures 11, 12). This included simulations that incorporated a period of elevated branching rates (50% increase) and a tenfold increase in extinction rate. However, the accuracy of the likelihood ratio test decreased when diversification rates varied through time and also included a period during which extinction was much greater than speciation rate, in which case the BDC model was erroneously rejected in favour of the incompatible rates model for 61–86% of replicates (Table 1, Supplementary Table 8). Importantly, these cases resulted in

erroneous rejection of the BDC model in favour of the independent rates model. This suggests that our approach to model testing is conservative, in that model violations, which are likely to occur in empirical data sets, will tend to artificially decrease support for the BDC model, rather than increase it. We relaxed the assumption of constant rates by implementing a Bayesian skyline version of the BDC model in which phylogenetic and fossil-based rates of speciation and extinction can vary over time (see Methods). We demonstrate using empirical data that accounting for rate variation can improve support for the BDC.

Finally, we assessed the impact of cryptic speciation (here indicating a speciation event not accompanied by recognisable phenotypic change and therefore unobserved in the fossil record[35,50]) on the support for our model. As the proportion of cryptic speciation increases, support for the compatible rates model decreased most frequently in favour of the equal rates model (Table 1, Supplementary Table 9, Supplementary Figures 13, 14). This is expected since undetected speciation events in the fossil record will remove the signal of additional cladogenetic and anagenetic speciation events that occur under the BDC model. In some cases, phylogenetic rate estimates may exceed stratigraphic estimates, which is in conflict with Eqs. (4)–(6), in which case the incompatible rates model was preferred.

Taken together, the results indicate that our modelling approach correctly identifies the impact of different speciation and extinction modes on diversification rates estimated using fossils and phylogenetic data. In particular, even when the discrepancies between data sets are large (e.g. $\lambda = 0.2$, $\mu = 0.16$ versus $\lambda^\star = 0.5$, $\mu^\star = 0.46$), our results show that the test is able to identify support for the BDC process. However, the results also highlight several factors that can reduce the power of the test and are important to consider when applying the test to empirical data sets.

**Empirical support for the BDC model**. To establish empirical support for the BDC model, we analysed fossil occurrence data and dated phylogenetic trees of nine plant and animal clades,

within Bayesian and maximum likelihood frameworks (see Methods). The clades are heterogeneous in terms of temporal range, size, taxon sampling, as well as their evolutionary history and ecology (Supplementary Table 14).

Phylogenetic and stratigraphic rate estimates differ substantially in most clades (Fig. 3). These differences are statistically significant in six out of nine clades, reinforcing the observation that inconsistencies between the two data types are ubiquitous among empirical data sets (Fig. 3, Supplementary Tables 10−12). Of the six data sets displaying rate discrepancies, we found that four clades conform to the expectations of the compatible rates model. Thus, although stratigraphic and phylogenetic estimates of speciation and extinction rates in these clades showed large discrepancies (e.g. 3- to 18-fold rate differences in Feliformia, Fig. 3b), these can be attributed to the occurrence of bifurcating and anagenetic speciation events, without the need to invoke any potential biases in the data.

Although the exact extent of different speciation modes cannot be inferred based on our analyses, we can use the properties of the BDC model to assess the prevalence of different speciation modes (Supplementary Table 11). For the three clades that show support for the equal rates model (Ursidae, Sphenisciformes, Canidae), we can conclude that budding was the prevalent mode of speciation, and that neither anagenesis nor bifurcation have contributed substantially to species diversification within these groups. Among the four clades that show support for the BDC model with compatible rates, our estimates indicate that anagenetic speciation was as important as budding speciation in Feliformia and Cervidae ($\lambda^{\star} - 2\lambda \approx 0$) (Fig. 4). In contrast, anagenetic speciation likely exceeded budding speciation in Bovidae and Cetacea ($\lambda^{\star} - 2\lambda > 0$). Finally, we can obtain posterior estimates of the sum of bifurcation and anagenetic rates of speciation ($\lambda_a + \lambda\beta = \lambda^{\star} - \lambda$), as shown in Supplementary Figure 16.

The BDC model was rejected in two data sets (ferns and corals), indicating that rate differences may not be entirely explained by different speciation modes. These data sets are characterised by much longer evolutionary histories than the other clades (i.e. hundreds of millions of years) and exhibited the greatest amount of temporal heterogeneity in both speciation and extinction rates (Supplementary Table 13, Supplementary Figure 15), which can result in spurious rejection of a constant rate BDC model (Table 1, Supplementary Table 7). We therefore reanalysed these data sets using a skyline implementation of the model, which allows for rate variation across different intervals (see Methods). Relaxing the assumption of constant rates resulted in strong support for the BDC model among ferns within each of seven time slices used in the analysis (Supplementary Table 12, Supplementary Figure 17). Both fossil and phylogenetic data supported significant rate heterogeneity through time (Fig. 5). As expected for a genus level data set, we found evidence that budding significantly exceeds anagenetic origination throughout most of the diversification history of the group (Fig. 5a).

Thus, among the nine empirical data sets tested here, only corals show evidence for a discrepancy between phylogenetic and palaeontological estimates of diversification rates, which cannot be reconciled under the BDC model (Supplementary Table 12). This likely reflects difficulties in the identification of taxonomic lineages in corals using morphological data (the only option for the ancient fossil record) in comparison with genetic and genomic data[51]. The phylogenetic data also appeared to have limited signal regarding the early diversification history of the clade, as reflected by the large uncertainty in rate estimates (Supplementary Figure 18). Taxonomic incongruences between modern and fossil data are likely to introduce conflicts favouring the incompatible rates model, as we have demonstrated in simulations incorporating cryptic speciation.

**Reconciling palaeontological and phylogenetic evidence.** There has been an intense, recent effort to integrate data from the fields of palaeontology and phylogenetics[18,52–55]. This includes not only advances in methods used to estimate diversification rates[13,46,56], but also models for estimating divergence times, phylogenetic relationships[54,57], and modes of phenotypic evolution[53,58]. Birth−death processes are fundamental to almost all methodological developments in this area of research, and consequently the definition of the parameters that underlie these models is extremely important[36,38].

Speciation and extinction rates inferred from fossil and phylogenetic data are usually given equivalent definitions, that is, the expected number of speciation or extinction events per lineage per unit of time[5,27,28,59]. We show that interpreting these quantities as equivalent parameters requires making the assumption that all species have been generated through a budding process. In fact, in a phylogenetic framework, $\lambda$ should be defined as a rate of branching, while $\mu$ is the rate of lineage extinction without replacement. Conversely, in fossil-based estimates, speciation and extinction rates quantify the pace at which (morpho)species originate and go extinct, regardless of the speciation mode or if extinction terminates a lineage or not.

The results of our empirical analyses suggest that, while our model may not be sufficient to fully capture the differences between fossil and phylogenetic evidence, the impact of alternative modes of speciation may play a previously overlooked role. Our findings also offer an explanation for striking differences in estimates of average species longevity, which are much shorter for fossil data relative to phylogenetic data[16], in which extinction with replacement is unaccounted for (see also ref. [33]). Understanding the conceptual differences between parameters that have previously been referred to using equivalent terms is a crucial step towards an improved integration of different data sources. Thus, the formulation of a BDC process has implications for (i) interpreting speciation and extinction rates estimated using different data sets, (ii) defining predictions about the differences between phylogenetic and fossil-based inferences, and (iii) establishing a coherent framework to simultaneously analyse phylogenetic and fossil data.

Evidently, several factors can generate incompatibility between palaeontological and phylogenetic parameter estimates, and we demonstrated some of these here, e.g. substantial rate variation, non-uniform sampling and cryptic speciation. Furthermore, speciation and extinction rates may be age-dependent[16,33,60], which may be an interesting aspect for future exploration of our model. Other processes are also of potential importance, such as speciation through hybridisation and reticulation, although their integration into evolutionary models is not straightforward[44]. Biased sampling of fossil and phylogenetic data, taxonomic inconsistencies, and dating errors will all contribute to disparities among data sets, and have each been defended as important aspects in the development of integrative models in palaeobiology. We have shown here that part of the incongruences observed between fossil and phylogenetic estimates can be attributed to a simple yet crucial conceptual difference in the meaning of speciation and extinction rates.

**Implications of the BDC model for understanding speciation.** A wide range of perspectives exist regarding the definition and the nature of species and the speciation process. For example, the evolutionary species concept tends to consider all speciation (and therefore branching) events as budding[40], while the Hennigian species concept tends to consider all speciation events as bifurcating[61]. Meanwhile, morphospecies that arose through anagenesis (i.e. speciation via replacement) are only accepted as true

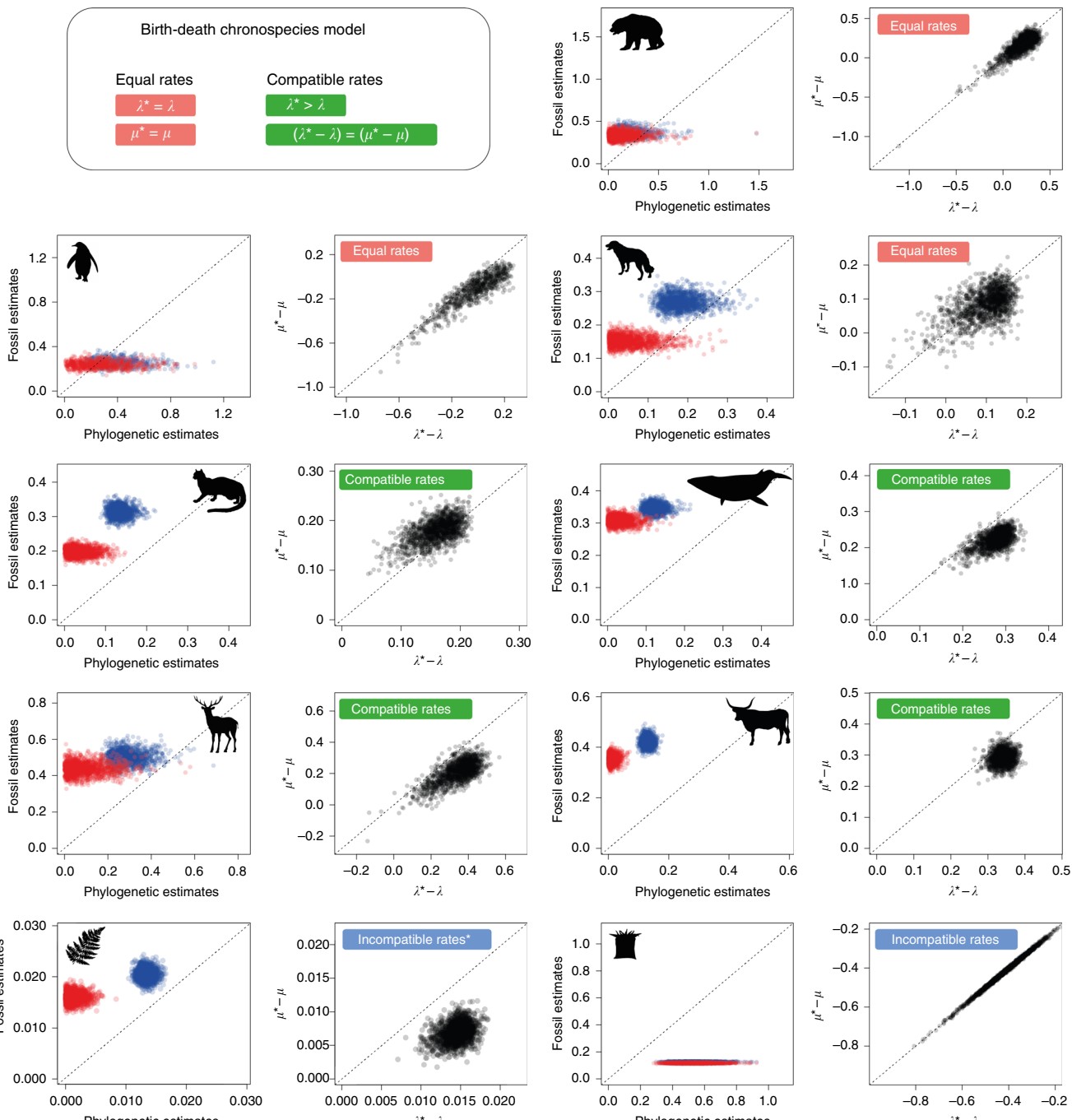

**Fig. 3** Results from a Bayesian analysis of fossil and phylogenetic data for nine clades. Posterior samples of speciation rates (in blue) and extinction rates (in red) jointly inferred from the two data types are plotted against one another; posterior samples of the two terms of Eq. (5) are shown in black. The results shown here are based on the assumption of constant rates through time. The clades include (from top to bottom): Ursidae, Sphenisciformes, Canidae, Feliformia, Cetacea, Cervidae, Bovidae, ferns and allies, and Scleractinia (see also Methods; animal silhouettes from www.phylopic.org; fern silhouette from www.publicdomainpictures.net). Although the analyses were run assuming independent rates ($\lambda$, $\mu$, $\lambda^\star$, $\mu^\star$), their joint posterior samples were used to assess which model (equal rates, compatible rates, or incompatible rates) best fit the data. The best model, indicated by the labels in the plots, is based on whether the posterior samples conform to the properties of the models (summarised in the top left panel) using a 99% threshold for significance. Consistent results were also obtained in a maximum likelihood framework (Table 1). The equal rates model is supported in three data sets, i.e. stratigraphic and phylogenetic rates are not significantly different. Stratigraphic and phylogenetic speciation and extinction rates are significantly different in four data sets, but compatible with the expectations of the BDC model. Finally, origination and extinction rates are significantly different in two data sets and the discrepancy cannot be explained by different speciation modes. However, the fern data set (bottom left) supported a BDC model after accounting for rate variation through time (Fig. 5)

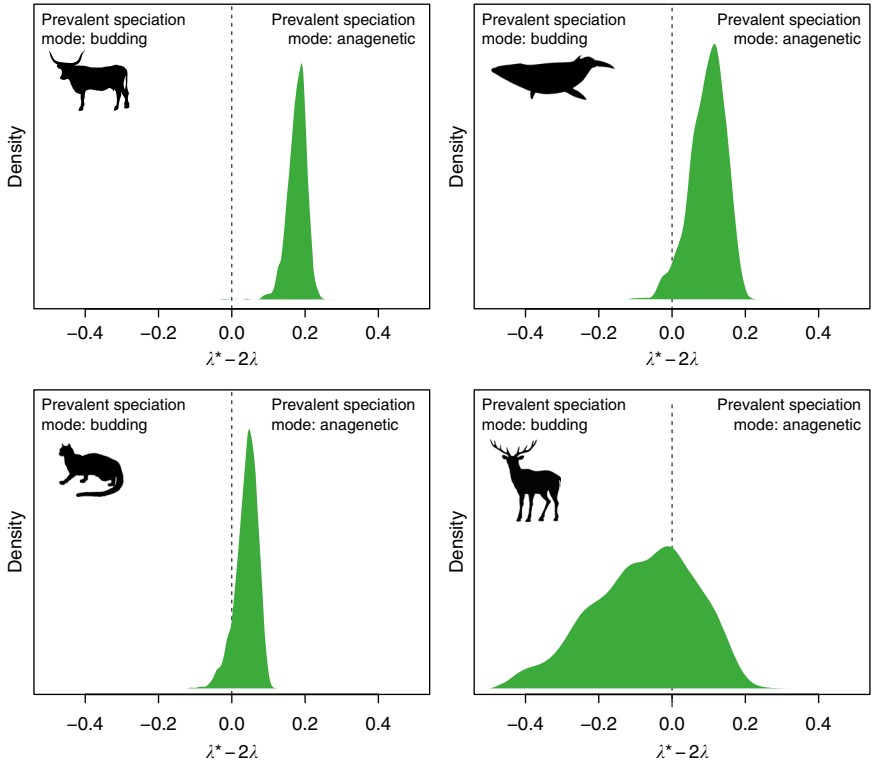

**Fig. 4** Prevalent mode of speciation inferred under the BDC model. Although the exact contribution of different speciation modes cannot be quantified under the BDC model, the joint analysis of fossil and phylogenetic data is informative about the relative importance of budding versus anagenetic speciation. The four clades shown here (Bovidae, Cetacea, Feliformia, Cervidae) show support for diversification under the compatible rates model, meaning stratigraphic and phylogenetic estimates of speciation and extinction rates are different, but can be explained by differences in speciation mode (Fig. 3). Based on the properties of the BDC model, positive values of $\lambda^* - 2\lambda$ (posterior distributions from a joint Bayesian analysis shown as density plots) indicate that the rate of anagenetic speciation exceeds the rate of budding speciation, whereas the opposite is true for negative values

species by some authors[42], on the basis that this process reflects phenotypic evolution along lineages rather than speciation per se[38,39].

Here, we have considered species in terms of the fundamental biological units used to obtain estimates of macroevolutionary parameters[44]. The description of species in the fossil record relies ultimately on phenotypic traits and we perceive changes in the pace of turnover of novel traits, or morphospecies, to reflect something of biological significance, regardless of whether these changes are associated with branching events of either type or trait turnover without branching. The combined outcome of different processes generating morphospecies will be reflected in the taxonomy of extinct taxa and therefore in the estimates of speciation and extinction rates. As we have demonstrated, these estimates will be distinct from estimates obtained from molecular phylogenies, and the discrepancies are actually informative about the speciation process under the BDC model.

The relative role of different speciation modes in the evolution of species remains largely unknown, but is likely to vary across taxonomic groups, as well as geographic and environmental contexts[62]. Previous work has found evidence of budding being the most prevalent mode among marine invertebrates[38,49], while evidence from plants suggests that anagenesis is the predominant mode[43], and conflicting evidence has been found among foraminifera[48,63]. However, previous work has not taken advantage of the distinction between diversification rates estimated from fossils and phylogenies. Our joint analysis of fossil and phylogenetic data provided empirical evidence of budding speciation being the most prevalent process in some clades (Supplementary Table 11, Fig. 5), but also instances

supporting a substantial contribution from anagenetic speciation (while the contribution of bifurcation remains elusive; Fig. 4). By formalising the definition of the parameters that are estimated from palaeontological versus phylogenetic data and the relationship between them, we have created a new opportunity with which to approach the topic of species evolution in the fossil record.

In this paper, we fitted the BDC model treating phylogenies and fossils as independent of one another, even though both data sets were generated by the same evolutionary process. Future methodological developments should aim to infer the four parameters of the BDC while explicitly taking into account the inter-dependencies between phylogenies and fossils[46,55], therefore directly quantifying the contributions of all three speciation modes.

**The sixth law of palaeobiology**. Recently, C. R. Marshall[15] proposed five palaeobiological laws essential for understanding the evolution of the living world, which have been broadly under-appreciated by neontologists. These laws emphasise the role of extinction in evolution, and the article joins many others in questioning the reliability of diversification rates estimated from phylogenetic trees[20,25], despite a substantial body of theoretical work to the contrary[10,23,26]. However, the fact remains that the fossil record documents historically high levels of species diversity that are not detected from phylogenetic trees. We have demonstrated that speciation and extinction rates inferred from palaeontological and phylogenetic data are expected to differ a priori, even in the absence of any biases, simply because they

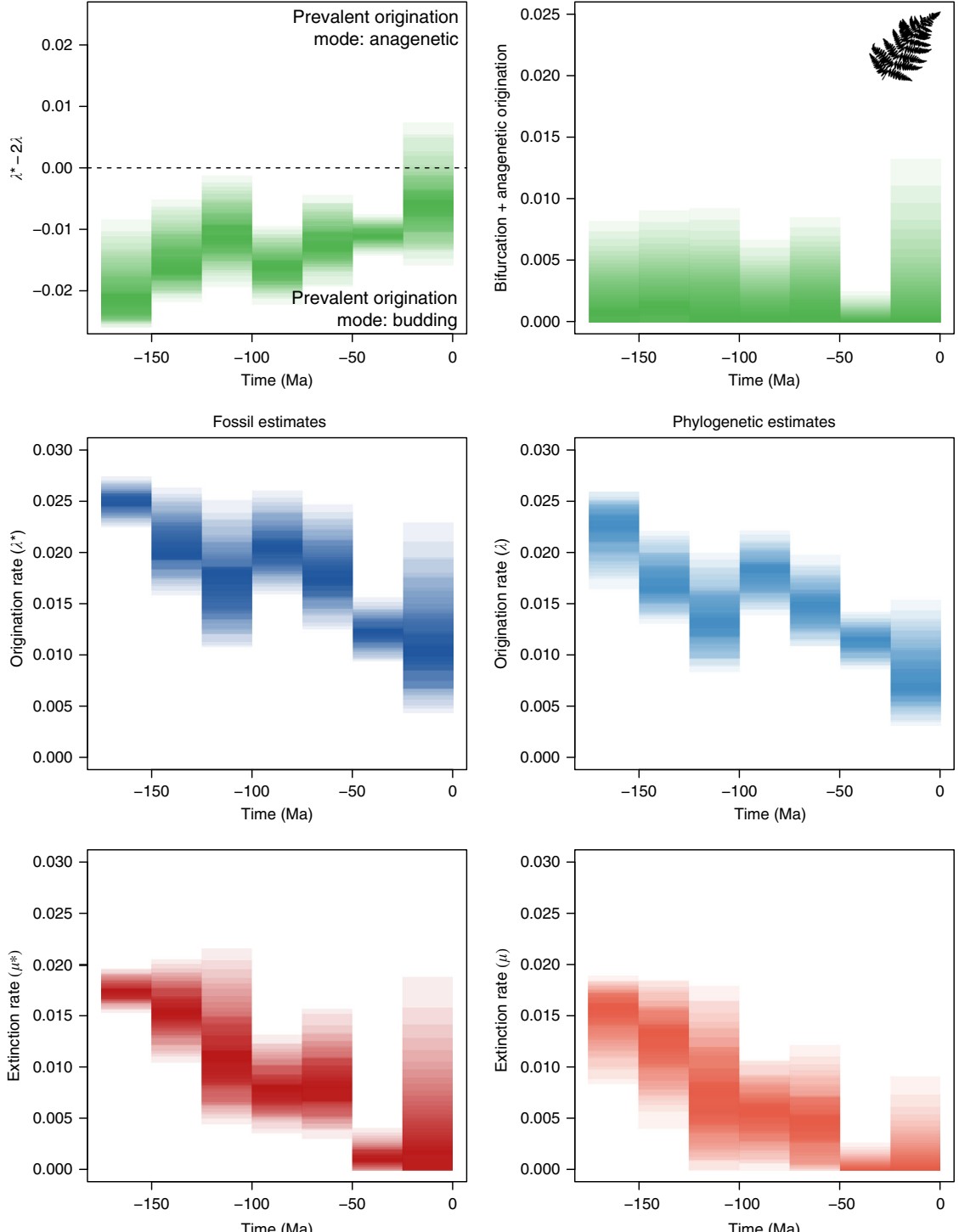

**Fig. 5** Analysis of the fern data set under the BDC skyline model. Origination and extinction rates were jointly inferred within each time bin from the fossil record and phylogenetic tree, under the constraints imposed by the BDC model. Prevalent modes of origination (in green) indicate that, as expected for a genus level data set, budding was more important than anagenetic origination, and that there is a low but non-zero rate of bifurcation and anagenesis. Origination and extinction rates inferred from fossils and from the phylogeny of ferns show a substantial amount of variation through time with a general tendency to decrease over time, although the net diversification remains positive in most of the time bins

measure different quantities. We also showed that a phylogenetic extinction rate of zero does not imply that species are immortal, since it ignores extinction associated with replacement, either through cladogenesis or anagenesis. Fully reconciling the discrepancies between phylogeny and fossil-based estimates of diversification rates therefore requires a better understanding of

the contribution of different speciation modes to the evolution (and description) of species in the fossil record, and how these processes relate to reconstructed phylogenies.

We propose a sixth law of palaeobiology that recognises the effects of different speciation modes on the estimation and interpretation of diversification rates obtained from

palaeontological and phylogenetic data. This law is given by Eqs. (1) and (2), which explicitly define the parameters associated with the processes of cladogenesis via budding or bifurcation and anagenesis, and their relationship to the diversification rates estimated from phylogenies and stratigraphic ranges. Our model illustrates that differences between fossil and phylogenetic estimates of speciation and extinction are expected and ultimately informative about the prevalent mode of speciation. The predictions of the sixth law are supported by the numerical and empirical results presented in this study, and may explain numerous other contrasting findings between phylogenetic and fossil estimates. Understanding and explicitly modelling the differences between phylogenetic and fossil species concepts should be the basis of future attempts to integrate the two data types.

## Methods

**Simulations**. To validate the expected relationship between $\lambda$, $\mu$ and $\lambda^\star$, $\mu^\star$, we used simulations incorporating phylogenetic branching processes and multiple modes of (morpho)species evolution. First we simulated constant rate birth−death trees using the R package TreeSim[64]. Three sets of 100 tree replicates were simulated with variable turnover: with branching rate $\lambda = 0.2$ and branch extinction rate $\mu = 0.02$, 0.1, or 0.16. We simulated trees conditioning on the number of extant tips, $n = 200$. The expected origin time for each set of trees is 32, 52 and 108 time units. Discrete chronospecies units were modelled by combining three alternative modes of speciation (cladogenesis via budding, cladogenesis via bifurcation and anagenesis) using the R package FossilSim (https://github.com/fossilsim/fossilsim). In a given tree, each branching event (or node) represents a budding speciation event with probability $1 - \beta$; thus speciation via bifurcation occurs with probability $\beta$, and anagenetic speciation occurs along each branch with rate $\lambda_a$. For each set of trees we generated species units using three alternative parameter combinations: $\beta = 0$ and $\lambda_a = 0$, $\beta = 0.5$ and $\lambda_a = 0.04$, or $\beta = 0.7$ and $\lambda_a = 0.16$. Note that when $\beta = 0$ and $\lambda_a = 0$, all new species occur via budding and under these circumstances we expect $\lambda = \lambda^\star$ and $\mu = \mu^\star$. Extant species phylogenies were generated by pruning all extinct taxa from the simulated trees. Stratigraphic range data were generated using the times of origination and extinction of the chronospecies simulated under the BDC speciation model.

**Budding versus anagenetic speciation**. To demonstrate that the properties of the birth−death chronospecies model can be used to determine the relative contribution of budding versus anagenetic speciation, we generated data sets with $\beta$ fixed to 0.5 and $\lambda_a$ set to 0.1, 0.2, 0.3 ($\lambda = 0.2$). We expect $\lambda^\star - 2\lambda$ to be negative when budding speciation exceeds anagenetic speciation, and positive when anagenetic speciation exceeds budding speciation (see 'Mathematical exploration of the BDC model' for further details of model expectations; Supplementary Table 2, Supplementary Figure 2).

**Incomplete species sampling**. Incomplete sampling affects many empirical phylogenetic trees and arguably all fossil data sets, erasing a proportion of the speciation and extinction events. To examine the impact of uniformly missing data on palaeontological diversification rate estimates and on support for the birth−death chronospecies model, we excluded each stratigraphic range from the initial simulated data sets prior to analysis with probability 0.1, 0.5 and 0.9 (Supplementary Table 3, Supplementary Figures 3, 4). To examine the impact of non-uniformly missing data, we excluded entirely extinct ranges only with probability 0.1, 0.5, 0.9 (Supplementary Table 5, Supplementary Figures 7, 8), and in a separate set of analysis we excluded extant ranges only with probability 0.1, 0.5, 0.9 (Supplementary Table 6, Supplementary Figures 9, 10). Note that in these experiments, we did not remove any species from the extant species phylogenies. However, removing species from the extant species phylogenies is expected to have a similar impact on the results, i.e. uniformly missing extant tips will increase the variance in parameter estimates but not necessarily reduce support for the compatible rates model, while non-uniformly missing tips may increase variance and reduce support for the compatible rates model.

**Fossil sampling**. To assess the impact of fossil sampling on our model testing approach, we simulated fossil occurrence data under a Poisson process with sampling rate $\psi = 0.5$. Under this model the probability of sampling a given range will be a function of range duration, i.e. shorter ranges are less likely to be sampled. In the resulting fossil occurrence data sets, the start and end of ranges will be represented by first and last appearances, and thus will underestimate the total range duration. To obtain estimates of the true range durations (i.e. true origination and extinction times) we analysed the fossil occurrence data using the program PyRate (see section 'Analysis of empirical data' for further details). Estimated origination and extinction times were then used as input to obtain maximum likelihood estimates of $\lambda^\star$ and $\mu^\star$ for model testing.

**Rate heterogeneity**. Temporal variation in diversification rates is also common across clades but is not explicitly accounted for in our model. To assess the impact of temporal rate variation on the relationship between $\lambda$, $\mu$ and $\lambda^\star$, $\mu^\star$ we simulated three sets of trees under the diversification rate shift model implemented in TreeSim. To incorporate an episode of high diversification, we assigned $\lambda = 0.3$ and $\mu = 0$ to the interval 10−20 time units. Otherwise, all other parameters settings used to generate trees were implemented as above ($\lambda = 0.2$, $\mu = 0.02$, 0.1, or 0.16). This resulted in three sets of tree replicates with an expected origin time of 26, 34 and 52 time units. Second, we generated trees with an episode of high diversification followed by a large increase in extinction equal to speciation in the last interval. $\lambda = 0.2$ and $\mu = 0.02$ prior to 25 time units, $\lambda = 0.3$ and $\mu = 0$ during the interval 15−25 and $\lambda = \mu = 0.2$ during the interval 0−15; expected origin time = 40. Third, we generated trees under the scenario in which extinction rate in the final interval was much greater than speciation (i.e. the clade is in decline). $\lambda = 0.3$ and $\mu = 0.02$ prior to 25 time units, $\lambda = 0.2$ and $\mu = 0.01$ during the interval 15−25 and $\lambda = 0.2$ and $\mu = 0.3$ during the interval 0−15; expected origin time = 45. In all cases stratigraphic ranges were generated from the simulated trees as above, with different combinations of $\beta$ and $\lambda_a$. Speciation and extinction rates were estimated from completely sampled data sets (Supplementary Table 7, Supplementary Figures 11, 12).

**Cryptic speciation**. We define cryptic speciation as a speciation event that does not involve any recognisable phenotypic change, a case that is not accounted for in the birth−death chronospecies model. Cryptic lineages are likely to be common in clades with a fragmentary fossil record or when phenotypic changes have low probability of being preserved (e.g. traits associated with soft tissues). To examine the impact of cryptic speciation on palaeontological diversification rate estimates and on support for the birth−death chronospecies model, we generated three data sets incorporating cryptic species. At each speciation event (budding, bifurcating or anagenetic) we assigned a probability $\kappa = 0.2$, 0.5, or 0.9 of the event being cryptic, in which case the new species becomes undistinguishable from its parent species in the fossil range data set. Note this does not affect the data used to estimate rates from phylogenies. Speciation and extinction rates were estimated from completely sampled data sets (Supplementary Table 9, Supplementary Figures 13, 14).

**Incompatible rates**. To establish that our test correctly rejects support for the BDC model when phylogenetic and palaeontological rates are generated under the independent rates model, we simulated data sets of trees and ranges with independent sets of parameters. We generated two sets of 100 trees under a constant rate birth−death model, in each replicate $i$ sampling the parameter values from uniform distributions: $\lambda_i^1 \sim \mathcal{U}(0.1, 1.5)$ and $\mu_i^1 \sim \mathcal{U}(0, \lambda_i)$ and $\lambda_i^2 \sim \mathcal{U}(0.1, 1.5)$ and $\mu_i^2 \sim \mathcal{U}(0, \lambda_i)$. One set of trees was used to generate extant phylogenies while the second set was used to generate data sets of ranges. We then analysed sequential pairs of trees and ranges from each independent set using maximum likelihood and tested among the equal, compatible, and incompatible rate models. Anagenesis and bifurcating speciation were not incorporated into these simulations (i.e. $\lambda_a = 0$ and $\beta = 0$).

**Parameter inference**. Phylogenies of extant species and temporal range data were used to calculate speciation and extinction rates using phylogenetic and palaeontological approaches, respectively. Maximum likelihood estimates of $\lambda$ and $\mu$ were estimated using the birth−death model described by Stadler[65] (Eq. (2)), which requires information about the origin time ($t_0$), i.e. the stem age of the clade, and the age of internal nodes ($\mathcal{T} = [t_1, ..., t_{n-1}]$) in a phylogeny of $n$ extant species,

$$f(\mathcal{T}|t_0, \lambda, \mu, \rho) = \frac{p_1(t_0)}{1 - p_0(t_0)} \prod_{i=1}^{n-1} \lambda p_1(t_i) \qquad (7)$$

with,

$$p_0(t) = 1 - \frac{\rho(\lambda - \mu)}{\rho\lambda + (\lambda(1-\rho) - \mu)e^{-(\lambda-\mu)t}},$$
$$p_1(t) = \frac{\rho(\lambda-\mu)^2 e^{-(\lambda-\mu)t}}{(\rho\lambda + (\lambda(1-\rho)-\mu)e^{-(\lambda-\mu)t})^2}. \qquad (8)$$

The parameter $\rho$ is the probability of including an extant species into the phylogeny. As we included all extant species in all analyses of simulated data, we set $\rho = 1$ throughout. We constrained the diversification rate to be positive (i.e. $\lambda > \mu$) in all analyses. We emphasise that this approach does not use any information about the delimitation of chronospecies.

Maximum likelihood estimates of $\lambda^\star$ and $\mu^\star$ based on stratigraphic range data ($\mathcal{R}$) were calculated using the birth−death model described in refs [13,28], based on the number of birth events ($B$) and death events ($D$), and on the sum of range durations ($S$),

$$P(\mathcal{R}|\lambda^\star, \mu^\star) \propto \lambda^{\star B} \mu^{\star D} e^{-(\lambda^\star + \mu^\star)S}. \qquad (9)$$

In all cases we assumed complete sampling, either of extant or extinct species, unless otherwise specified.

The results using data simulated under complete sampling are shown in Supplementary Table 1 and Fig. 2, Supplementary Figure 14.

**Likelihood ratio test.** To assess support for the birth−death chronospecies model under the above simulation conditions and to establish when support for the model is expected to break down, we implemented a likelihood ratio test, comparing three alternative models. Each model describes the distribution of phylogenies and the distribution of stratigraphic ranges with likelihood function:

$$P(\mathcal{T}|\lambda,\mu), \qquad P(\mathcal{R}|\lambda^*,\mu^*), \qquad (10)$$

where $P(\mathcal{T}|\lambda,\mu)$ is defined by Eq. (7) and $P(\mathcal{R}|\lambda^*,\mu^*)$ by Eq. (8). We obtain the maximum likelihood estimates for $\lambda$, $\mu$ by maximising $P(\mathcal{T}|\lambda,\mu,)$, and for $\lambda^*$, $\mu^*$ by maximising $P(\mathcal{R}|\lambda^*,\mu^*)$. We determine the model best describing our data by using a likelihood ratio test. We approximate the joint probability

$$P(\mathcal{T},\mathcal{R}|\lambda,\mu,\lambda^*,\mu^*) \approx P(\mathcal{T}|\lambda,\mu,)P(\mathcal{R}|\lambda^*,\mu^*), \qquad (11)$$

and then perform a likelihood ratio test. Thus we calculate LR = $2(\log \text{ML}_{H1} - \log \text{ML}_{H0})$, where $\text{ML}_H = \max_{\lambda,\mu,\lambda^*,\mu^*} P_H(\mathcal{T},\mathcal{R}|\lambda,\mu,\lambda^*,\mu^*)$. Since the three models differ by the constraints imposed on parameters $\lambda$, $\mu$, $\lambda^*$, $\mu^*$ we use the following rules to compare the statistical fit of alternative models.

In the Equal rates (H0) versus Compatible rates (H1) comparison the constraint $\lambda^* = \lambda$ is relaxed to $\lambda^* \geq \lambda$. Thus, LR = $2(\log \text{ML}_{CRM} - \log \text{ML}_{ERM})$ is a mixture of a $\chi^2$ distribution with one degree of freedom (i.e. a $\chi_1^2$) and a dirac delta distribution (i.e. all probability mass is at 0) because $\lambda^* - \lambda$ is restricted to zero in the null hypothesis and therefore lies on the border of possible values $\lambda^* - \lambda > 0$ under the alternative hypothesis[66]. Thus, we reject equal rates at a level $\alpha$ if $P_{\chi_1^2}(X > \text{LR}) > 2\alpha$. E.g., at a level $\alpha = 0.05$, we reject the equal rates model if LR > 2.71.

In the Compatible rates (H0) versus Incompatible rates (H1) comparison the constraint $\lambda^* \geq \lambda$ is relaxed to $\lambda^* > 0$, and $\mu^* = \lambda^* - \lambda + \mu$ to $\mu^* \geq 0$. However, we only test the constraint $\mu^*$ as we could not find the distribution of the likelihood ratio statistic taking into account both constraints. For this we use a $\chi_1^2$ distribution for the distribution of LR = $2(\log \text{ML}_{IRM} - \log \text{ML}_{CRM})$, since in log-space (i.e. considering $\log(\mu^*)$ instead of $\mu^*$) the constraint translates to a fixed parameter versus a parameter taking any value in $(-\infty, \infty)$. Generally, the real type-1 error of our test should be slightly higher than $\alpha$, as we only consider the change in the constraint on $\mu^*$ and not on $\lambda^*$. However, this should increase the power compared to a test at the level $\alpha$. The test should therefore be conservative by slightly under-estimating the threshold yielding $\alpha = 0.05$, thus favouring the rejection of the compatible rates model.

In the Equal rates (H0) versus Incompatible rates (H1) comparison the constraint $\lambda^* = \lambda$ is relaxed to $\lambda^* > 0$, and $\mu^* = \lambda^* - \lambda + \mu$ to $\mu^* \geq 0$. We assume a $\chi_2^2$ distribution for LR = $2(\log \text{ML}_{IRM} - \log \text{ML}_{ERM})$, since for the parameters in log-space, our setting translates to two fixed parameters versus both parameters taking any value in $(-\infty, \infty)$.

We note that Table 1 (main text) summarises the empirically determined type-1 errors for our likelihood ratio test in some simulation scenarios, revealing that our approximation in calculating the joint probability (Eq. (10)) and our rejection procedure produces the expected results.

Model parameters were estimated using maximum likelihood optimisation for combined data sets of phylogenies and stratigraphic ranges. All maximum likelihood optimisations were repeated five times using different initial values to reduce the probability of finding a local optima, and the results with the highest likelihood score were selected. We performed model testing using the likelihood ratio tests described above. In our tests, we used two thresholds for statistical significance set to 0.95 and 0.99.

**Analysis of empirical data.** Using empirical data available for nine clades, we assessed whether there was significant incongruence between the diversification rates estimated from different data sets and, if so, whether any incongruences could be explained by the BDC model. We obtained empirical data sets from the following sources: (i−iii) Feliformia, Canidae, Ursidae, fossil occurrence data from ref. [67], phylogenies from ref. [68]; (iv) Cetacea, phylogeny from ref. [69], fossil occurrence data retrieved from the Paleobiology Database (https://paleobiodb.org/) using the R library paleobioDB[70]; (v) Ferns and allies (genus level), phylogeny and fossil data from ref. [71]; (vi−vii) Bovidae and Cervidae, phylogeny and fossil data from ref. [17]; (viii) Scleractinia, phylogeny and fossil data from ref. [72]; (ix) Spheniciformes, phylogeny from ref. [73] and fossil occurrence data retrieved from the Paleobiology Database, as above. Fossil data comprise fossil occurrence times for each extinct and extant species with a known fossil record. Phylogenetic data consisted of dated phylogenetic trees of extant taxa.

To incorporate uncertainties associated with the fossil record, in addition to the maximum likelihood inference described above, we analysed the empirical data within a Bayesian framework, using a new implementation of the program PyRate[74] developed for this study. This allowed us to analyse fossil and phylogenetic data to jointly estimate: (i) the times of origination and extinction of each fossil lineage, (ii) preservation rates through time, (iii) fossil-based speciation and extinction rates ($\lambda^*$, $\mu^*$), and (iv) phylogenetic speciation and extinction rates ($\lambda$, $\mu$). We estimated independent preservation rates within each geological epoch assuming a time-variable Poisson process and constant rate birth−death models

for fossils and phylogeny (with independent parameters $\lambda^*$, $\mu^*$ and $\lambda$, $\mu$). The birth−death processes were estimated using the likelihood functions described in Eq. (10), and we used PyRate's default gamma priors on the birth−death rates (with shape and rate equal to 1.1). Because several of the empirical phylogenies did not include 100% of the known extant taxa, we corrected for missing lineages by setting the $\rho$ parameter (see Eq. (7)) as specified in Supplementary Table 14. Since for the empirical phylogenies we do not know the age of the origin and we conditioned the process on the age of the crown, rather than the origin[65]. Although the stratigraphic and phylogenetic rates here are assumed to be fully independent parameters (as in the incompatible rates model), we used their joint posterior distributions sampled using Markov Chain Monte Carlo (MCMC) to assess the support for each model. We ran 20 million MCMC iterations to obtain posterior estimates of the parameters and used posterior samples of $\lambda^*$, $\mu^*$, $\lambda$, and $\mu$ to verify the conditions predicted by the birth−death chronospecies model (Eqs. (4)−(6)).

The estimated times of origination and extinction and the phylogenies of extant taxa were used to test the equal, compatible and incompatible rates models under the maximum likelihood framework described above. Under the Bayesian framework, the equal rates model was selected if 0 was included in the 95 or 99% credible intervals of $\lambda^* - \lambda$ and $\mu^* - \mu$. The compatible rates model was preferred if stratigraphic and phylogenetic rates were different, but 0 was included in the 95% (or 99%) credible interval of $(\lambda^* - \lambda) - (\mu^* - \mu)$ and if $P(\lambda^* \geq \lambda) > 0.05$ (or 0.01). The incompatible rates model was preferred if none of the conditions above were met. We then re-analysed the data sets for which the BDC model was preferred, after constraining the parameter values sampled by the MCMC based on the assumptions of the equal or compatible rates models. We used the posterior samples of $\lambda$ and $\lambda^*$ to make inferences about the prevalence of different speciation modes (Fig. 4, Supplementary Table 11).

We used the estimated times of origination and extinction for fossil lineages to infer the amount of rate variation from fossil data only. We used the reversible-jump MCMC algorithm[75] implemented in PyRate to infer the number and temporal placement of rate shifts and to obtain the marginal rates through time[13,76]. We then computed the ratio between the greatest and the smallest marginal rates (independently for speciation and extinction) as a measure of the magnitude of rate variation in the data (Supplementary Figure 15).

In addition, we implemented a BDC skyline model in which speciation and extinction rates may vary across predefined time bins. This extension, only available within the Bayesian implementation, is based on the birth−death models with rate shifts described in ref. [22] for phylogenetic data and in ref. [13] for fossil data. In the analysis of the fern and coral data sets, time bins were set to 25 Myr in length starting from time 0 (the present) going back to 150 Ma, with the earliest bin extending from 150 Ma to the time of origin of the clade (>400 Ma). This partition scheme was selected to guarantee sufficient statistical power to estimate speciation and extinction rates with both phylogenetic and fossil data. As with the constant rate model, we first ran the analysis under the assumption of independent rates to assess whether the BDC model was supported. We then ran another analysis on the fern data under the BDC model to estimate compatible phylogenetic and fossil rates and the prevalence of different speciation modes.

**Empirical simulations.** To demonstrate that we should expect to find support for the birth−death chronospecies (compatible rates) model given the scale of our empirical data (in terms of phylogenetic and stratigraphic range data size, age and diversification rates), we simulated data sets under the BDC model, based on the parameters obtained for each of the nine data sets (Supplementary Table 14). First, we estimated speciation and extinction rates from the empirical phylogenies $(\hat{\lambda}, \hat{\mu})$, and used them to simulate trees after setting the number of terminal tips to the present diversity of each clade. Simulated trees were then used to (1) simulate fossil ranges and (2) simulate phylogenies of extant taxa. We simulated stratigraphic range data using $\beta = 0.5$ (this value was chosen arbitrarily) and for each simulated tree, $\lambda_a$ was iteratively increased from 0.1 until the number of simulated ranges was $\geq$ the empirical number of ranges. Prior to analysis the range data were uniformly pruned to match the number of empirical ranges. We generated phylogenies of extant taxa by pruning all extinct tips. For clades with incomplete taxon sampling ($\rho < 1$) we additionally removed a random set of taxa to reach the observed sampling fraction. Model testing using maximum likelihood was performed as described above. These simulations demonstrate that we should expect to find overall support for the birth−death chronospecies model given the scale and parameters of our empirical data sets (or similar data sets; Supplementary Table 14).

**Code availability.** The maximum likelihood implementation of the BDC model (constant rate model only) is available in the R package fbdR (https://github.com/rachelwarnock/fbdR). The Bayesian implementation, which includes the BDC skyline model, is available in the latest version of PyRate[74] (https://github.com/dsilvestro/PyRate). The code and scripts used in this study are also available at https://doi.org/10.5281/zenodo.1471499 (http://zenodo.org/record/1471499)[77] accompanied by readme files explaining their use.

## Data availability

All the simulated and empirical data (fossil occurrences and phylogenetic trees) presented and analysed in this study are available online at https://doi.org/10.5281/zenodo.1471499 (http://zenodo.org/record/1471499)[77].

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

## Acknowledgements
We thank David Bapst and Charles Marshall for providing valuable feedback on the manuscript. In addition, we thank Ziheng Yang, Andreas Steingötter and the ETH Seminar for Statistics group for providing advice on model testing and Carl Simpson for providing data. D.S. received funding from the Swedish Research Council (2015-04748) and from the Swedish Foundation for Strategic Research. R.C.M.W. was funded by the ETH Zürich Postdoctoral Fellowship and Marie Curie Actions for People COFUND programme. T.S. is supported in part by the European Research Council under the Seventh Framework Programme of the European Commission (PhyPD: grant agreement number 335529). A.G. was funded by the Bioprotection Research Centre. Part of the analyses were run at the high-performance computing centre Vital-IT of the Swiss Institute of Bioinformatics (Lausanne, Switzerland).

## Author contributions
D.S., R.C.M.W., A.G., and T.S. designed the study and developed the methods. D.S. and R.C.M.W. wrote the manuscript with contributions from all authors. R.C.M.W. implemented the maximum likelihood tests and ran the simulations. D.S. implemented the Bayesian version of the BDC model and analysed the empirical data sets.

## Additional information

**Competing interests:** The authors declare no competing interests.

