## [Peer Review File · Nature Communications]

Reviewers' Comments:

Reviewer #1:

Remarks to the Author:

Silvestro & Warnock et al. propose that the major reason for a discrepancy in evolutionary rate estimates between paleontological and neontological approaches. The treatment takes a mathematical and simulation approach to investigate the question, illustrating performance on high-quality empirical datasets. I think the manuscript makes a valuable contribution to help reconcile differences where none should be, thus bringing us closer to a more inclusive evolutionary biology, which should (in my opinion) be a goal for a journal like Nature Communications. The text is predominantly well-written, although I have a few suggestions for where it may become easier to follow.

Pages 15 and 16 contain an apparent contradiction: 312 states that "in reality, the true mode of speciation will almost never be known" whereas 320 suggests that this new model "may even allow us to estimate the prevalence of different speciation modes". I see how these two statements aren't mutually exclusive, but this seems awkward. I appreciate the authors do not want to open Pandora's Box on species concepts (72-74) but isn't one interpretation of these results is that modes 2 and 3 are human constructs to accommodate incomplete data that are never present in reality?

Statements like this would seem appropriate in the provocative "sixth law of palaeobiology" section that closes the manuscript. On that subject, state what you believe this new law to be in unequivocal terms - the subheading demands it.

Returning to the problematic modes 2 & 3 (163), which biases results the most? Or is their interaction greater than the sum of the two parts?

Another point that could be clarified is the differences between speciation and extinction from 275. Re-emphasising the need to look beyond net diversification rate could be re-emphasised here (citing the Foote reference that is currently 50). Another way of describing the statistical link here is hazard rates (the approach taken in ref. 32), in which the instantaneous chance of speciation (or extinction) can be written in terms of the conditional rate assuming that no budding or (no death) has occurred previously. This can be seen in the statistical distributions, where the "time to speciation" distribution will be less than the "time to extinction" distribution as it needs to be "reset" to 0 after each budding speciation. The more these two distributions overlap, the harder it will be to disentangle speciation and extinction components of evolutionary rates.

I disagree with the fossil-based different statement on line 277, unless there is not a phylogeny attached to the calculations, in which one might question the study of evolutionary rates without clear knowledge of the ancestor.

Minor suggestions

p2, "that is necessary" -> "that is able"

29: isn't it a biological signature, rather than fossil record per se. I know we don't see it in neontological systems at the moment but we may be able to do if we document divergences for long enough.

45: outliers is a formal (or at least recognised) statistical term; is this justified here?

98: Don't you need to cite Gingerich here somewhere?

Gingerich, P. D. 1979. The stratophenetic approach to phylogeny reconstruction in vertebrate paleontology. In J. Cracraft and N. Eldredge (eds.), *Phylogenetic Analysis and Paleontology*, Columbia University Press, New York, pp. 41-77. PDF or Request PDF/reprint 72

or

Gingerich, P. D. 1990. Stratophenetics. In D. E. G. Briggs and P. R. Crowther (eds.), *Palaeobiology: A Synthesis*, Blackwell Scientific Publications, Oxford, pp. 437-442. PDF or Request PDF/reprint 210

149: statistically significant?

165: The pseudospeciation and pseudoextinction point from 307 could come here

258: replace "paleontological research" with "these groups"; a lot of paleontological research is not macroevolutionary and plenty that occurs at the species level.

Reviewer #2:

Remarks to the Author:

see the attached pdf.

Reviewer #3:

Remarks to the Author:

Silvestro et al have produced an extremely interesting manuscript. The idea that speciation (or perhaps better called origination when in the context of fossils) rate inferred from phylogenetic trees and fossil data are different in nature is interesting (but perhaps not that surprising). This is excellently explained in the text and shown in their Figure 1.

As far as I am concerned the paper is technically excellent – the correction based on Λ^* and μ^* is very elegant and simple. And I think it represents an important addition to the speciation and extinction literature. However, I am not convinced about the suitability of this manuscript for *Nature Communication*. I think it would be far better placed in a more specialist journal such as *Systematic Biology* or *Methods in Ecology and Evolution*. This is by no means a reflection on the quality of the work – as I have said I think it is excellent and well done.

The reason I am not convinced it is suitable for publication in the current *Journal* is that I think the model lacks biological realism - Λ^* and μ^* are not really biological parameters, rather they are a statistical correction factor. But we are left wondering what this means in terms of real biology. The authors realize this themselves. They say "we aim to understand if different modes of speciation and extinction, irrespective of whether they reflect true biological processes or taxonomic practices, can explain and even predict the incongruences observed between phylogenetic and palaeontological estimates of macroevolutionary rates" and "Here, we have refrained from making explicit statements about the true mechanisms of speciation, or the prevalence of different speciation modes (budding, bifurcation, anagenesis)." I find this a shame as I am sure with some more investigation they could actually do this.

Linked to this – but also extremely important in its own right – is the fact that the model is almost always rejected in favour of the incompatible model when varying rates exist through time. This highlights that the correction might not have any biologically interpretable value. But perhaps more important is that it shows this model will often not be useful as much research in this area (including a great deal by the authors here) demonstrate that speciation and extinction rates routinely vary through time. In fact I would say that many think varying rates through time is ubiquitous.

For these reasons, I find the final section before the methods – *The Sixth Law of Palaeobiology* - so

farfetched and fanciful I would recommend removing that from any future submission of this work to any journal.

Minor point: the new model here is inconstantly named. Is sometimes called "the independent rates" model – perhaps that was a previous name.

In sort I think this paper is interesting and will invigorate research in this area – but owing to the lack of biological interpretation I cannot recommend publication in Nature Communications.

Reviewer #1 (Remarks to the Author):

Silvestro & Warnock et al. propose that the major reason for a discrepancy in evolutionary rate estimates between paleontological and neontological approaches. The treatment takes a mathematical and simulation approach to investigate the question, illustrating performance on high-quality empirical datasets. I think the manuscript makes a valuable contribution to help reconcile differences where none should be, thus bringing us closer to a more inclusive evolutionary biology, which should (in my opinion) be a goal for a journal like Nature Communications. The text is predominantly well-written, although I have a few suggestions for where it may become easier to follow.

We thank the reviewer for this assessment.

Pages 15 and 16 contain an apparent contradiction: 312 states that "in reality, the true mode of speciation will almost never be known" whereas 320 suggests that this new model "may even allows us to estimate the prevalence of different speciation modes". I see how these two statements aren't mutually exclusive, but this seems awkward. I appreciate the authors do not want to open Pandora's Box on species concepts (72-74) but isn't one interpretation of these results is that modes 2 and 3 are human constructs to accommodate incomplete data that are never present in reality?

In revising our manuscript, we have expanded significantly the exploration of the BDC model and the use of its properties to make inferences about the prevalence of different speciation modes. While we still refrain from taking a strong position on the nature of a "true species" (as this is clearly not the objective of our study), we now show that there are at least some aspects about speciation mode that can be inferred from the result of a joint analysis of fossil and phylogenetic data, namely the relative contribution of anagenetic versus budding speciation. We now discuss the interpretation of model parameters with respect to speciation mode in more detail and the results of the joint inference in two new paragraphs in the Results (l. 91-98 and l. 214-232) and Discussion (l. 368-398) and demonstrate the interpretation of speciation modes empirically (l. 298-308; Figure 4; Table S11; Fig. S16).

Statements like this would seems appropriate in the provocative "sixth law of palaeobiology" section that closes the manuscript. On that subject, state what you believe this new law to be in unequivocal terms - the subheading demands it.

We thank the reviewer for raising this point. We have revised this section of the manuscript and now state explicitly what we propose as the sixth law of palaeobiology, and describe the expectations under this law (l. 421-440).

Returning to the problematic modes 2 & 3 (163), which biases results the most? Or is their interaction greater than the sum of the two parts?

Anagenetic speciation and bifurcation contribute similarly to determining the discrepancy between fossil-based and phylogenetic estimates of speciation and extinction rates. We now state this explicitly in the manuscript (l. 201-203).

Another point that could be clarified is the differences between speciation and extinction from 275. Re-emphasizing the need to look beyond net diversification rate could be re-emphasized here (citing the Foote reference that is currently 50).

We added this reference. However, we think that a more thorough discussion of the use of net diversification rates in traditional phylogenetic studies (which is mostly linked to the higher statistical power to estimate it compared with the speciation and extinction rates individually) would be a bit distracting here, since in our manuscript we only show speciation and extinction rates.

Another way of describing the statistical link here is hazard rates (the approach taken in ref. 32), in which the instantaneous chance of speciation (or extinction) can be written in terms of the conditional rate assuming that no budding or (no death) has occurred previously. This can be seen in the statistical distributions, where the "time to speciation" distribution will be less than the "time to extinction" distribution as it needs to be "reset" to 0 after each budding speciation. The more these two distributions overlap, the harder it will be to disentangle speciation and extinction components of evolutionary rates.

We are not entirely sure how hazard functions can be incorporated into our BDC model. The instantaneous extinction rate is μ^* under our model. This rate is constant throughout a species lifetime, unless the species itself changes its λ , μ , β or λ_a . We now however mention that age-dependent variation would be an interesting aspect to be explored in future studies (line 356-358).

I disagree with the fossil-based different statement on line 277, unless there is not a phylogeny attached to the calculations, in which one might question the study of evolutionary rates without clear knowledge of the ancestor.

We have rephrased the sentence to clarify that the speciation and extinction rates estimated from fossil data do not differentiate between speciation modes and between extinction of a terminal lineage and extinction with replacement (l. 337).

Minor suggestions

p2, "that is necessary" -> "that is able"

Done.

29: isn't it a biological signature, rather than fossil record per se. I know we don't see it in neontological systems at the moment but we may be able to do if we document divergences for long enough.

What we mean here is that anagenetic and bifurcating events are not visible (do not leave a signature) in a phylogeny of extant taxa, where only budding events are observed.

45: outliers is a formal (or at least recognised) statistical term; is this justified here?

Rephrased (l. 62)

98: Don't you need to cite Gingerich here somewhere?

Gingerich, P. D. 1979. The stratophenetic approach to phylogeny reconstruction in vertebrate paleontology. In J. Cracraft and N. Eldredge (eds.), *Phylogenetic Analysis and Paleontology*, Columbia University Press, New York, pp. 41-77. PDF or Request PDF/reprint 72

or

Gingerich, P. D. 1990. Stratophenetics. In D. E. G. Briggs and P. R. Crowther (eds.), *Palaeobiology: A Synthesis*, Blackwell Scientific Publications, Oxford, pp. 437-442. PDF or Request PDF/reprint 210

Excellent suggestion. Gingerich 1979 is now cited in both the Results and Discussion sections.

149: statistically significant?

Changed to “substantial” (l. 177)

165: The pseudospeciation and pseudoextinction point from 307 could come here

We now no longer use the terms “pseudospeciation” or “pseudoextinction” in the revised text.

258: replace "paleontological research" with "these groups"; a lot of paleontological research is not macroevolutionary and plenty that is occurs at the species level.

Done (l. 317).

Reviewer #2 (Remarks to the Author):

Summary

The authors propose a simple explanation for discrepancies between estimates of speciation and extinction rates that come from phylogenetic studies and from fossil occurrence data. The explanation is very elegant, both scientifically and mathematically, and is based on the idea that lumping multiple modes of speciation leads to incompatibility of model parameter definitions in the phylogenetic and fossil occurrence models. The authors also provide empirical support of their hypothesis about the source of discrepancies between the two data sources for speciation and extinction rates.

Positives

1. The mathematical and scientific expositions are very clear and not overly technical.
2. Silvestro and co-authors carefully think about statistical identifiability of model parameters and about how empirical data support their conclusions.

We thank the reviewer for this assessment.

Negatives

1. I may have missed it, but did the authors study the effect of missing data in fossil occurrence data? The likelihood that the authors use for these data assumes completely observed birth-death process in continuous time. How sensitive the authors' results to this assumption being true? Is this easy to address via simulations?

We agree with the reviewer that this point was not fully developed in the previous version of the manuscript. The reviewer is correct in noting that the likelihood function utilized for fossil ranges does not correct explicitly correct for missing data. We have shown in previous papers that the effects of missing data on the accuracy of the birth-death rate estimates is

- 1) negligible when sampled lineages are random subset of the total (Silvestro et al. 2014 doi: 10.1093/sysbio/syu006; Fig. S1)
- 2) very limited when data incompleteness derives from low preservation rates (Silvestro et al. 2014 Syst Biol Fig. 4, see also Silvestro et al. 2018 doi:10.1101/316992)

Following the reviewer's advice, we have nevertheless run additional simulations to more thoroughly assess the effects of incomplete fossil data on the birth-death-chronospecies model (l. 250-252, 456-458; Supplementary text l. 49-57; Table S4; Figs. S5-6). In addition to testing the effect of randomly incomplete taxon sampling (which was already included in the previous version of the manuscript), we simulated datasets in which fossil occurrence were sampled from a Poisson process. Under this scenario, short-lived lineages are less likely to be preserved in the fossil record than long-lived ranges, thus generating a non-random incomplete sampling. The new results confirm the robustness of our test to the biases introduced by the preservation process (Table 1).

2. In the supplementary materials the authors say “We performed model testing using likelihood ratio tests, based on chi-squared distributions with degrees of freedom equal to the difference in number of free parameters between each model.” This could be problematic, because the likelihood ratio tests have different asymptotic properties when the constraints of the null model form singularities. For example, constraining a model parameter on the boundary of its domain leads to the LRT distribution being a mixture of chi-squared distributions (Chernoff’s work). The authors should either justify their choice of the null distribution for LRT (either theoretically or via simulations) or change their null distribution to account for singularities.

We thank the reviewer for raising this point, which we have addressed by changing the reference distribution in our likelihood ratio tests. We now describe more explicitly the differences in parameter constraints between models (l. 160-173) and have substantially updated the relevant section of the Supplementary Materials in which the tests are described (paragraph: Likelihood ratio test; l. 115-153). We repeated all simulations based on the new significance thresholds and updated all the figures and tables associated with them.

Suggestions

1. In the discussions the authors say that they hope to jointly estimate all 4 parameters of the parameters of their BDS model. I am curious why the authors have done this in the current study. The authors are already doing something very similar when they perform likelihood ratio tests. Namely, they work with the joint likelihood:

$$P(T, R \mid \lambda, \mu, \lambda^*, \mu^*) = P(T \mid \lambda, \mu)P(R \mid \lambda^*, \mu^*), \text{ where } P(R \mid \lambda^*, \mu^*) \propto (\lambda^*)^B(\mu^*)De^{-(\lambda^* + \mu^*)S}.$$

Why not work with the joint likelihood expressed in terms of the BDS parameters

$$P(T, R \mid \lambda, \mu, \beta, \lambda a) = P(T \mid \lambda, \mu)P(R \mid \lambda, \mu, \beta, \lambda a),$$

$$\text{where } P(R \mid \lambda, \mu, \beta, \lambda a) \propto [\lambda(1-\beta) + 2\lambda\beta + \lambda a]^B[\lambda\beta + \lambda a + \mu]De^{-(\lambda + 2\lambda\beta + 2\lambda a + \mu)S}?$$

While we could in principle sample the BDC parameters from the joint likelihood, Equation 3 in the manuscript shows that the parameters $\beta, \lambda a$ cannot be directly estimated, yielding infinite, equally-likely solutions (point B, p. 8 in our manuscript).

We highlight again that the above formulation (just like ours) treats the fossils and tree as independent data points – but they are correlated (i.e. sampled from the same underlying complete tree). Thus, we need tools to consider the tree with fossil sampling along lineages for estimating all four parameters, which will potentially allow to estimate all four parameters. This is now more clearly stated in the discussion (l. 399-403).

Reproducibility and software availability

1. fbdR and fossilsim packages need a minimal example in the Readme file. For fossilsim, an easy solution would be to move or copy one of the examples from the example directory to

the Readme file. The fbdR package lacks examples entirely, so the need for a better Readme is even more urgent.

Both packages have standard R documentation and examples associated with all functions. The README file for both packages has been updated to reflect this and includes more information about available documentation and examples.

Minor Comments

1. "We call this process on four parameters the BDC process." → "We call this process with four parameters the BDC process."

Done.

2. Fonts in Figures 2 and 3 need to be enlarged for better readability.

Done.

3. In the supplement, the birth-death process likelihood has a typo: the minus sign should be outside the parentheses.

Fixed.

Reviewer #3 (Remarks to the Author):

Silvestro et al have produced an extremely interesting manuscript. The idea that speciation (or perhaps better called origination when in the context of fossils) rate inferred from phylogenetic trees and fossil data are different in nature is interesting (but perhaps not that surprising). This is excellently explained in the text and shown in their Figure 1.

We thank the reviewer for this assessment.

As far as I am concerned the paper is technically excellent – the correction based on Λ^* and μ^* is very elegant and simple. And I think it represents an important addition to the speciation and extinction literature. However, I am not convinced about the suitability of this manuscript for Nature Communication. I think it would be far better placed in a more specialist journal such as Systematic Biology or Methods in Ecology and Evolution. This is by no means a reflection on the quality of the work – as I have said I think it is excellent and well done.

The reason I am not convinced it is suitable for publication in the current Journal is that I think the model lacks biological realism - Λ^* and μ^* are not really biological parameters, rather they are a statistical correction factor. But we are left wondering what this means in terms of real biology. The authors realize this themselves. They say “we aim to understand if different modes of speciation and extinction, irrespective of whether they reflect true biological processes or taxonomic practices, can explain and even predict the incongruences observed between phylogenetic and palaeontological estimates of macroevolutionary rates” and “Here, we have refrained from making explicit statements about the true mechanisms of speciation, or the prevalence of different speciation modes (budding, bifurcation, anagenesis).” I find this a shame as I am sure with some more investigation they could actually do this.

We thank the reviewer for this comment, which made us rethink about the (lack of) emphasis we had given to the biological interpretation of the parameters in our model. In our manuscript we had tried to avoid opening the Pandora box regarding species concepts (as also pointed out by Reviewer 1) and this rationale led us to downplay -perhaps too much- the biological implications of the BDC model. Although we still think a full discussion about what constitutes a “real species” is outside of the scope of this paper, we agree with the reviewer that a better effort should be made to explain the biological interpretations of the parameters of the BDC model.

We do not consider \$\lambda^*\$ and \$\mu^*\$ as merely statistical corrections factors. In fact, they do quantify the rate at which entities, identified on the basis of morphology, originate and disappear through time. These entities, often referred to as ‘morphospecies’, may not be “real species” but there is no universal consensus about what a “real species” is. However, they do have extremely valuable biological meaning when we consider that, for instance, events such as mass extinctions or ancient radiations are only known and described based on originations and extinctions of morphospecies.

In our revised manuscript, we have elaborated more on the biological interpretation of the parameters described in our model, referring to several seminal studies from the literature providing the theoretical framework on which we built the different speciation modes (l. 44-49; l. 91-98). Importantly, we expanded on a previously neglected property of our model: the combined analysis of fossil and phylogenetic data under the BDC model is informative of the prevalence of different speciation modes (l. 67-72; l. 213-232). Although we did have a small paragraph about this property in our initial submission, we had not sufficiently emphasized its interpretation and application in empirical data sets. In addition, we worked out a way to compute the interval of plausible values for the rate of anagenetic speciation, which we present in our revised manuscript (l. 223). We can now also examine the contribution of anagenetic versus budding speciation and the sum of anagenetic and bifurcating speciation (l. 223), while the contribution of bifurcating speciation remains elusive. We now exploit the properties of the BDC model to examine the prevalence of different speciation modes for the seven empirical clades that support the BDC model. Our findings are reported in a new figure (Fig. 4) and in the Results (l. 297-306; see also Table S11 and Fig. S16). Finally, we now provide a more thorough discussion of how the BDC model can help us understanding speciation in a dedicated section of the Discussion (l. 368-398).

Linked to this – but also extremely important in its own right – is the fact that the model is almost always rejected in favour of the incompatible model when varying rates exist through time. This highlights that the correction might not have any biologically interpretable value. But perhaps more important is that it shows this model will often not be useful as much research in this area (including a great deal by the authors here) demonstrate that speciation and extinction rates routinely vary through time. In fact I would say that many think varying rates through time is ubiquitous.

We agree that rate variation is likely to be almost ubiquitous in empirical datasets, which is why we have explored in detail (1) the effects of rate variation on BDC estimates with simulated datasets (Table 1, S7, S8) and (2) the amount of rate variation in the empirical datasets analysed here (Fig. S15). In our simulations, we note that the BDC model is erroneously rejected very frequently only in the scenario with the highest level of rate variation, whereas at more moderate levels of rate variation model testing is still accurate at the 0.99 significance threshold (Table 1; 96 and 94% accuracy).

Our simulations demonstrate the need to extend our model into a birth-death skyline framework, which is mathematically possible but non-trivial. The simulations incorporating rate variation were included to demonstrate that when rates vary – which we agree will be the norm – the model may be erroneously rejected, but this does not mean that the model does not apply. In fact, since rate variation tends to reduce the statistical support for the BDC model in favour of the incompatible rate model, our results from empirical data provides even stronger evidence for the importance of different speciation modes in contributing to the diversity observed in the fossil record. We have now extended and clarified these points in the manuscript (l. 262-266).

For these reasons, I find the final section before the methods – The Sixth Law of Palaeobiology - so farfetched and fanciful I would recommend removing that from any

future submission of this work to any journal.

We respectfully disagree with the reviewer on this point, especially in the light of the implications that our method has on two fronts: (1) the estimation of different speciation modes (l. 368-398) and (2) the establishment of the theoretical basis for a coherent framework to analyse fossil and phylogenetic data. The previous 5 laws of palaeobiology were defined to facilitate the unification of neontological and palaeontological data. Our new model, which underpins the sixth law, complements this effort. Here, we demonstrate that unification cannot fully be achieved without considering more explicitly how the terms “speciation” and “extinction” apply to different datasets. The first 5 laws were also written in response to the observation that there are often huge and unexplained discrepancies between diversification rates estimated from phylogenies and the fossil record. These discrepancies are often erroneously attributed to our inability to estimate extinction from phylogenies. The new model provides a strong theoretical basis for the observed discrepancies, including directionality.

We have now thoroughly reworked the section (l. 421-440), providing a clearer definition of what we think of as the 6th law and why we think it provides a crucial step towards a full and coherent integration of neontological and paleontological evidence in understanding the dynamics of species diversification and extinction.

Minor point: the new model here is inconstantly named. Is sometimes called “the independent rates” model – perhaps that was a previous name.

We have polished the text to make the reference to our new model more consistent throughout the manuscript.

In sort I think this paper is interesting and will invigorate research in this area – but owing to the lack of biological interpretation I cannot recommend publication in Nature Communications.

We hope our revised version of the manuscript clarifies the wide-ranging implications of our model, which we think expand beyond the purely technical methodological progress contributing to a more general understanding of speciation and extinction. Following the reviewer’s advice we have extended the discussion of the biological implications of the BDC model and of how different speciation modes may shape diversification patterns. In light of these changes we believe our study will interest a broad community of evolutionary biologists working with neontological and/or palaeontological data.

Reviewers' Comments:

Reviewer #1:

Remarks to the Author:

I felt this manuscript contributes strongly to our understanding of how biodiversity evolves before the revisions, and I still think this now. I only have a few outstanding points:

On 248, presumably this "up to 90%" is contingent on having reasonably large sample sizes in the first instance? (443 implies $n=200$ for the birth-death simulations). How does this change at smaller sizes more routinely encountered in the palaeontological literature.

The point on 301 about the prevalence of budding speciation is more important than its current position. Mention in the abstract?

On 323, an interesting aspect of cryptic speciation is that it often is not uniformly distributed (as I presume is assumed here?). What would happen if all the cryptic diversity was contained within (say) half the species?

The age-dependency in branching rates (337-338, 356-358) was the "hazard rate" point I attempted to make in the rebuttal. I'll try again. The different species concepts modelled in ref 33 was achieved by different ways of slicing up lineages (as per your Figure 1), especially for extinction (bifurcation means extinction whereas budding does not). Figure 2 in ref 33 shows how bifurcation ("Hennigian") and budding ("persistent ancestry") show decreasing and increasing hazard rates, respectively (the dashed lines in panels a & b). A particularly interesting aspect to explore in the future is whether these statistical patterns on a particular clade emerge analytically.

Reviewer #2:

Remarks to the Author:

The authors addressed my concerns.

One minor comment: In the results section of the main manuscript, the authors say "likelihood test" when they mean "likelihood ratio test."

Reviewer #3:

Remarks to the Author:

The authors have taken my comments very seriously and have made changes to focus more on the biological interpretation of their results. I think this has improved the manuscript enormously and has changed my mind about its suitability for publication in Nature Communications.

The one issue I still have is associated with the simulations and the implementation in a skyline model – I still do not see how the results of the simulation are useful. I think they highlight a problem as I described in my last comments. This could be totally overcome and tested if the model was implemented in a skyline form. I understand that this is not trivial – but it is not an enormous amount of work, especially given the team of authors here – one could not imagine a group of people better able to do this! I think this inclusion of this would hugely improve the manuscript. In addition, I think it would massively increase the paper's biological impact and methodological half-life. With this addition I am sure it would make a great paper in Nature Communication.

Your revised manuscript entitled "Closing the gap between palaeontological and neontological speciation and extinction rate estimates" has now been seen by the original three referees. You will see from their comments below that the referees continue to find your work of interest; however, some points remain to be addressed. We are interested in the possibility of publishing your study in *Nature Communications*, but would like to consider your response to these concerns in the form of a revised manuscript before we make a final decision on publication.

Dear Editor,

We thank the reviewers for a positive and constructive assessment of our manuscript.

Once again, we have taken their suggestions very seriously. In particular, Reviewer 3 strongly motivated us to implement an extension of our birth-death chronospecies model that allows speciation and extinction rates to vary through time, i.e. a BDC skyline model. We applied this model to the empirical datasets that showed strong evidence of rate variation (ferns and corals). One exciting outcome of this new analysis is that it increases the empirical support for the BDC model – the model is now supported in 8 out of 9 datasets. We have addressed all other concerns of the reviewers and provide detailed responses below. We highlighted the changes in our revised manuscript using red font.

In light of these changes, we would be delighted if you would consider this version of our manuscript for publication in *Nature Communications* and look forward to hearing your response.

Daniele Silvestro on behalf of all co-authors.

Reviewer #1 (Remarks to the Author):

I felt this manuscript contributes strongly to our understanding of how biodiversity evolves before the revisions, and I still think this now. I only have a few outstanding points:

We thank again the reviewer for the positive and constructive comments.

On 248, presumably this "up to 90%" is contingent on having reasonably large sample sizes in the first instance? (443 implies $n=200$ for the birth-death simulations). How does this change at smaller sizes more routinely encountered in the palaeontological literature.

Yes, a reasonable number of lineages is necessary for diversification rate analyses (under any model) and removing 90% of the taxa from a clade of ten species would certainly not result in a reasonably sized dataset. We note that, in addition to simulations removing a total proportion of diversity (e.g. 90%, SI, lines 37-50), we performed simulations incorporating the fossil sampling process, using empirical estimates of fossil recovery (SI, lines 51-60), and simulations where the rates and the size of simulated datasets were based on our empirical data (SI, lines 194-209). Thus, we feel satisfied that our simulations cover a wide range of sampling scenarios that encompass reality.

The point on 301 about the prevalence of budding speciation is more important than its current position. Mention in the abstract?

It is challenging to get all of our main points across within the word limit of the abstract (= 150 words). However, we do state in the final sentence of the abstract that the model is informative about the underlying speciation process.

On 323, an interesting aspect of cryptic speciation is that it often is not uniformly distributed (as I presume is assumed here?). What would happen if all the cryptic diversity was contained within (say) half the species?

We agree that cryptic speciation is likely to be non-uniform across clades and although this was not incorporated into the current study, our current simulations indicate that this would lead to a decrease in support for the BDC model. Simulations show that both cryptic speciation and rate variation lead to decreased support for the model. Since non-uniform cryptic speciation would lead to artificial non-uniformity in speciation and extinction rates across lineages this would likely also reduce support for the model.

The age-dependency in branching rates (337-338, 356-358) was the "hazard rate" point I attempted to make in the rebuttal. I'll try again. The different species concepts modelled in ref 33 was achieved by different ways of slicing up lineages (as per your Figure 1), especially for extinction (bifurcation means extinction whereas budding does not). Figure 2 in ref 33 shows how bifurcation ("Hennigian") and budding ("persistent ancestry") show decreasing and increasing hazard rates, respectively (the dashed lines in panels a & b). A particularly interesting aspect to explore in the future is whether these statistical patterns on a particular clade emerge analytically.

While we assume that the parameters of the BDC do not depend on species age, we agree that this may be a simplistic assumption (as mentioned in the manuscript; lines 368-369). For the future, it would indeed be well-worth exploring the impact of species age-dependent parameters. However, that will be a major endeavor as we have shown in recent studies using the birth-death models without the chronospecies component (Hagen et al. Syst Biol 2015, Alexander et al. Syst Biol 2016, and Hagen et al. Syst Biol 2017).

Reviewer #2 (Remarks to the Author):

The authors addressed my concerns.

One minor comment: In the results section of the main manuscript, the authors say "likelihood test" when they mean "likelihood ratio test."

We thank the reviewer for catching this mistake and have corrected this.

Reviewer #3 (Remarks to the Author):

The authors have taken my comments very seriously and have made changes to focus more on the biological interpretation of their results. I think this has improved the manuscript enormously and has changed my mind about its suitability for publication in Nature Communications.

The one issue I still have is associated with the simulations and the implementation in a skyline model – I still do not see how the results of the simulation are useful. I think they

highlight a problem as I described in my last comments. This could be total overcome and tested if the model was implemented in a skyline form. I understand that this is not trivial – but it is not an enormous amount of work, especially given the team of authors here – one could not imagine a group of people better able to do this! I think this inclusion of this would hugely improve the manuscript. In addition, I think It would massively increase the papers biological impact and methodological half-life. With this addition I am sure it would make a great paper in Nature Communication.

We thank the reviewer for this encouragement. We have now implemented a Bayesian skyline version of the BDC model and run it on the two empirical data sets that were found to be incompatible with the BDC under constant rates – corals and ferns. The former remained incompatible demonstrating a genuine discrepancy between phylogenetic and paleontological signals, which we think could derive from a combination of factors, including sampling biases, errors in molecular dating, and taxonomic uncertainties. For the fern dataset on the other hand, we now find full support for the BDC model across 7 time slices, indicating that fossils and phylogenies converge to a compatible solution after accounting for temporal rate variation. The skyline version of the BDC model is now available in the latest version of the PyRate program (github.com/dsilvestro/PyRate) and the commands necessary to run this analyses are now provided in the Supporting Information (lines 12-13).

We agree with the reviewer that the inclusion of the BDC skyline model increases the applicability of our method on empirical datasets. We maintain, however, that a full implementation of the model incorporating model-testing (to determine the number and time of rate shifts) and accounting for the non-independence of phylogenetic and fossil data, will require a major effort and cannot be integrated in this study (although we hope it is something we can develop further in the future).

Reviewers' Comments:

Reviewer #3:

Remarks to the Author:

None